# CHARTS ARE NOT IMAGES:
# ON THE CHALLENGES OF SCIENTIFIC CHART EDITING

**Shawn Li[1], Ryan Rossi[2], Sungchul Kim[2], Sunav Choudhary[2], Franck Dernoncourt[2]**
**Puneet Mathur[2], Zhengzhong Tu[3], Yue Zhao[1]**
[1]University of Southern California, [2]Adobe Research, [3]Texas A&M University
`(li.li02, yue.z)@usc.edu`
`(ryrossi, sukim, schoudha, dernonco, puneetm)@adobe.com`
`tzz@tamu.edu`

## ABSTRACT

Generative models, such as diffusion and autoregressive approaches, have demonstrated impressive capabilities in editing natural images. However, applying these tools to scientific charts rests on a flawed assumption: a chart is not merely an arrangement of pixels but a visual representation of structured data governed by a graphical grammar. Consequently, chart editing is not a pixel-manipulation task but a structured transformation problem. To address this fundamental mismatch, we introduce *FigEdit*, a large-scale benchmark for scientific figure editing comprising over 30,000 samples. Grounded in real-world data, our benchmark is distinguished by its diversity, covering 10 distinct chart types and a rich vocabulary of complex editing instructions. The benchmark is organized into five distinct and progressively challenging tasks: single edits, multi edits, conversational edits, visual-guidance-based edits, and style transfer. Our evaluation of a range of state-of-the-art models on this benchmark reveals their poor performance on scientific figures, as they consistently fail to handle the underlying structured transformations required for valid edits. Furthermore, our analysis indicates that traditional evaluation metrics (e.g., SSIM, PSNR) have limitations in capturing the semantic correctness of chart edits. Our benchmark demonstrates the profound limitations of pixel-level manipulation and provides a robust foundation for developing and evaluating future structure-aware models. By releasing *FigEdit* (`https://github.com/adobe-research/figure-editing`), we aim to enable systematic progress in structure-aware figure editing, provide a common ground for fair comparison, and encourage future research on models that understand both the visual and semantic layers of scientific charts.

## 1 INTRODUCTION

Vision-language models (VLMs) have advanced rapidly, showing strong results in recognition, captioning, and instruction-following image editing (Radford et al., 2021; Schuhmann et al., 2022; Rombach et al., 2022; Brooks et al., 2023a; Zhang et al., 2023; Team et al., 2023; OpenAI, 2024; Chen et al., 2024b; Wang et al., 2024a; Lu et al., 2024; Liu et al., 2024b; Li et al., 2024a; Yao et al., 2024; Xu et al., 2024). Beyond natural images, chart editing focuses on the precise modification of charts and graphs from natural-language instructions, which is central to scientific communication and data analysis. Typical workflows include updating figures when upstream tables change, adapting layouts for publication, aligning styles across related plots, and converting encodings to highlight specific trends. In collaborative environments, edits often arrive as multi-turn requests with references to earlier messages, related figures, or localized visual cues. Such use cases require outputs that remain faithful to underlying data, consistent with visualization rules, and auditable for provenance (Belouadi et al., 2024). At the same time, instruction-tuning and dialogue-centric editing continue to expand the ability of modern systems to follow multi-turn control (Li et al., 2024d; Huang et al., 2024a; Ma et al., 2025; Wei et al., 2024b; Hahn et al., 2024; Deng et al., 2025; Zhang et al., 2025; Li et al., 2024c; 2025b; Shawn et al., 2025; Li et al., 2025a).

Despite these advances, figure editing differs fundamentally from natural image manipulation. A chart is the rendering of structured data through a graphical grammar, and valid edits are *structured transformations* on marks, scales, encodings, and legends rather than pixel changes. Instructions such as "add a bar for category *X* with value 42" require coherent updates to data schema and visual mappings, yet current models often treat them as visual rearrangements, producing outputs that appear plausible but violate semantics. This exposes a persistent problem–method mismatch: instruction-following editors and multi-turn generation systems (Brooks et al., 2023a; Zhang et al., 2023; Huang et al., 2023; Wang et al., 2024b) are optimized for perceptual alignment under open-ended goals, whereas figure editing is constrained by data fidelity and visualization rules. Models trained on web-scale natural images (Schuhmann et al., 2022; Radford et al., 2021) lack inductive bias to preserve value–encoding consistency, axis coherence, and legend integrity. While dialog-driven clarification (Andukuri et al., 2024; Chen et al., 2024a; Zelikman et al., 2024) or OCR augmentation (Rodriguez et al., 2023a;b) can mitigate ambiguity locally, they do not guarantee structure-preserving edits, leaving the core mismatch unresolved.

**Current approaches and benchmarks.** On the approach side, diffusion editors and multimodal LLMs have been extended to multi-turn control and retrieval-augmented interaction (Li et al., 2024d; Huang et al., 2024a; Ma et al., 2025; Wei et al., 2024b; Wang et al., 2025; Hahn et al., 2024; Deng et al., 2025; Li et al., 2026; Liu et al., 2024c; Taneja & Goel, 2025; Zhao et al., 2025b). Yet, these systems rarely operate on executable specifications or enforce semantic constraints, which makes them unsuitable for structured figure editing. On the benchmark side, prior chart-related datasets have mainly targeted captioning, QA, table extraction, or chart-to-code generation (Hsu et al., 2021; Kantharaj et al., 2022; Masry et al., 2023; Han et al., 2023; Zhang et al., 2024c; Xia et al., 2024; Shi et al., 2024a; Masry et al., 2024; Zhang et al., 2024b). As shown in Tab. 1, these resources leave several gaps. Some lack real underlying data altogether (e.g., Xia et al. 2024; Zhang et al. 2024d), reducing their grounding in authentic visualization workflows. Coverage of edit categories is also narrow: data-level updates, layout transformations, and style changes are often missing. Interactive scenarios such as visual guidance or style transfer are almost entirely absent, despite being common in real practice. Even the recent ChartEdit benchmark (Zhao et al., 2025a), while closer to editing, only partially spans instruction types and lacks paired figure outputs for direct comparison. Overall, existing benchmarks fall short of representing the breadth of figure editing and still depend heavily on pixel-level similarity metrics, which do not reflect semantic correctness. This highlights the need for a task-structured, semantics-aware, and scale-ready benchmark dedicated to figure editing.

**Our benchmark.** We introduce *FigEdit*, a large-scale benchmark for scientific chart editing with over 30,000 instances collected from realistic sources (Fig. 1). It spans 10 chart types and a diverse set of instructions, as summarized in Tab. 2, and is organized into five evaluation settings: single edits, multi edits, conversational edits, visual-guided edits, and style transfer edits. The benchmark also covers a wide range of operation categories, including data-centric edits, layout adjustments, style modifications, and text updates, detailed in Tab. 3. Unlike prior benchmarks that lack real data or paired chart outputs, *FigEdit* grounds edits in authentic charts and provides both charts and specification references. To address the absence of interactive scenarios, it includes conversational editing for multi-turn consistency, visual-guided editing with localized cues, and style transfer for cross-chart alignment. Finally, beyond SSIM and PSNR, *FigEdit* introduces semantics-aware evaluation that verifies transformations at the level of data and encodings, with executable targets or programmatic specifications where possible (Li et al., 2024b; Zhang et al., 2024a; Zheng et al., 2023; Wei et al., 2024a; Guo et al., 2024; Shi et al., 2024a). These design choices directly address the limitations of existing benchmarks and shift evaluation from pixel similarity toward semantic correctness in structured editing.

Our contributions are summarized as follows:

- **Problem formalization**: We define chart editing as a *structured transformation* task governed by a graphical grammar, clarifying required invariants such as data–encoding alignment, axis coherence, and legend integrity.

- **Task-structured benchmark**: We present *FigEdit*, a benchmark with 30K+ instances and 10 chart types, spanning single, multi, conversational, visual-guided, and style transfer with a diverse instruction set.

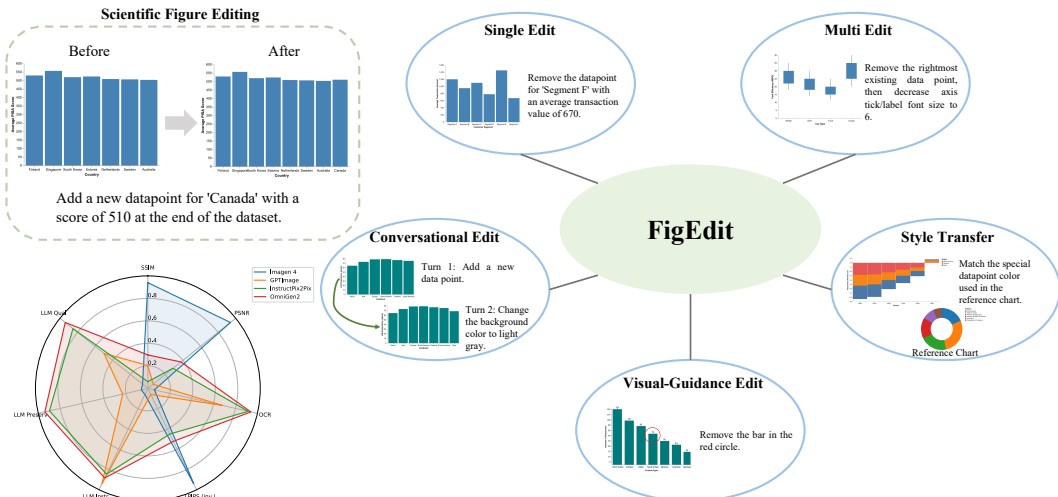

Figure 1: FigEdit benchmark. Top-left: an example figure illustrating the basic task. Bottom-left: a radar chart comparing model performance on single edit task, highlighting the benchmark's ability to reveal differences in editing capabilities. Right: taxonomy of the benchmark covering five tasks (single edit, multi edit, conversational edit, visual guidance, and style transfer).

- **Comprehensive study**: We systematically evaluate state-of-the-art editors and VLMs, showing that strong scores on pixel metrics do not imply correct structured edits, and analyze frequent failure modes.

## 2 RELATED WORK

**Text-to-Image Generation.** The rapid progress of diffusion-based models has revolutionized text-conditioned image generation, enabling results that are both high-fidelity and prompt-faithful (Ramesh et al., 2022; Rombach et al., 2022). ControlNet and related approaches expand controllability by incorporating structural or spatial priors (Zhang et al., 2023). Yet these advances have focused primarily on natural imagery. Scientific figures remain relatively neglected, despite their demand for symbolic precision, calibrated spatial relationships, and embedded textual fidelity. Evaluations show mainstream systems often fail in data accuracy and layout coherence for scientific use cases (Zhang et al., 2024b). In response, specialized methods such as OCR-aware generative frameworks (Rodriguez et al., 2023b) and programmatic vector-graphic synthesis (Belouadi et al., 2024) highlight the need for tailored solutions.

**Image Editing.** Instruction-based editing has evolved from GANs and encoder-based systems toward diffusion-driven methods, which better balance realism with semantic alignment. A survey by Huang et al. (2025) provides a comprehensive overview of this transition. Representative works include LEDITS++ (Brack et al., 2024), which extends text-driven editing to unconstrained transformations; Emu Edit (Sheynin et al., 2024), which integrates recognition for localized precision; and Liu et al. (Liu et al., 2024a), which probe attention mechanisms to preserve semantic fidelity. More recent works push toward interactivity and compositionality: SmartEdit (Huang et al., 2024b) employs multimodal LLMs to compose edits, ProxEdit (Han et al., 2024) stabilizes transformations without tuning, and DragDiffusion (Shi et al., 2024b) enables point-based manipulation. AnyEdit (Yu et al., 2025) exemplifies the broader trajectory toward unified, general-purpose editing frameworks.

**Scientific Chart Editing.** Unlike natural images, charts encode structured data, calibrated axes, and embedded text, requiring semantic consistency and readability throughout editing. While a broad literature addresses diffusion-based editing of natural scenes (Brooks et al., 2023a; Huang et al., 2024b; Han et al., 2024; Sheynin et al., 2024; Brack et al., 2024; Shi et al., 2024b; Yu et al., 2025; Huang et al., 2025), research specific to scientific figures is limited. ScImage investigates the limitations of multimodal LLMs for figure generation (Zhang et al., 2024b); AutomaTikZ explores text-to-vector generation under programmatic constraints (Belouadi et al., 2024); and ChartEdit formulates

Table 1: Comparison of our proposed benchmark with existing chart-related benchmarks. While prior benchmarks mainly target captioning, QA, or chart-to-code generation, they provide limited coverage of editing operations and interactive settings. *FigEdit* is the first benchmark designed for evaluation of figure editing, supporting diverse chart types, multiple instruction categories, and interactive scenarios such as visual guidance and style transfer edits.

| Name | Output Format | w/ Real Data | Diverse Types | Visual Guidance | Style Transfer | Editing Instruction | | | | |
|---|---|---|---|---|---|---|---|---|---|---|
| | | | | | | Data | Format | Layout | Style | Text |
| ChartCraft (Yan et al., 2024) | Json | ✗ | ✗ | ✗ | ✗ | ✓ | ✓ | ✓ | ✓ | ✗ |
| Plot2Code (Wu et al., 2024) | Code | ✓ | ✓ | ✗ | ✗ | ✗ | ✗ | ✗ | ✗ | ✗ |
| ChartX (Xia et al., 2024) | Code | ✗ | ✓ | ✗ | ✗ | ✗ | ✗ | ✗ | ✗ | ✗ |
| AcademiaChart (Zhang et al., 2024d) | Code | ✗ | ✗ | ✗ | ✗ | ✗ | ✗ | ✗ | ✗ | ✗ |
| ChartMimic (Shi et al., 2024a) | Code | ✓ | ✓ | ✗ | ✗ | ✓ | ✗ | ✗ | ✗ | ✗ |
| ChartEdit (Zhao et al., 2025a) | Code | ✓ | ✓ | ✗ | ✗ | ✓ | ✓ | ✓ | ✓ | ✓ |
| FigEdit (Ours) | Figure | ✓ | ✓ | ✓ | ✓ | ✓ | ✓ | ✓ | ✓ | ✓ |

Table 2: Benchmark data statistics across chart types and editing tasks. Each entry shows the number of instances per task, with subtotals by chart family and overall totals.

| Chart Type | Single Edit | Multi Edit | Conv. Edit | Style Transfer | Visual Guidance |
|---|---|---|---|---|---|
| Area | 1463 | 586 | 369 | 406 | 299 |
| Line | 1649 | 593 | 398 | 424 | 399 |
| Bar | 1800 | 600 | 375 | 410 | 398 |
| Stacked-bar | 1800 | 600 | 398 | 498 | 396 |
| Pie | 1000 | 600 | 398 | 200 | 397 |
| Donut | 1000 | 600 | 399 | 200 | 398 |
| **Other** | | | | | |
| Box | 1199 | 568 | 388 | 200 | 198 |
| Violin | 1000 | 500 | 300 | 200 | 150 |
| Scatter | 1398 | 598 | 324 | 400 | 322 |
| Dot | 1796 | 599 | 383 | 458 | 398 |
| **Totals** | **14105** | **6244** | **3732** | **3400** | **3355** |
| | | | | | *All tasks combined:* **30836** |

chart editing as a multimodal evaluation benchmark (Zhao et al., 2025a). A common limitation in existing work is reliance on intermediate code (e.g., matplotlib) as the target of modification. While this guarantees structural validity, it reduces evaluation to code executability and neglects perceptual quality and user-facing usability. Thus, the field lacks benchmarks that jointly measure instruction adherence, semantic fidelity, and visual clarity in an end-to-end setting, motivating figure editing as a distinct line of inquiry.

## 3 BENCHMARK

We introduce a *figure–centric* benchmark for scientific figure editing. Ground truth (GT) images are obtained by applying deterministic edit functions to Vega[1]/Vega–Lite[2] specifications and rendering the results. Evaluation is performed in image space. This design provides pixel-consistent supervision across atomic edits, one-shot composite edits, multi-turn conversations, figure edits with visual guidance, and figure edits with referenced figures, without depending on package-specific code.

### 3.1 FORMAL DEFINITION OF A CHART

A natural image $I$ can be viewed as a function mapping 2D coordinates to color values, $I : \mathbb{R}^2 \to \mathbb{R}^3$. In contrast, a chart is the rendered output of a structured specification. Formally, we define a deterministic renderer $R$ that maps a specification $\sigma \in \Sigma$ to an image $I \in \mathbb{R}^{H \times W \times 3}$:

$$I = R(\sigma). \tag{1}$$

---

[1]https://vega.github.io/

[2]https://vega.github.io/vega-lite/

Each specification $\sigma$ can be decomposed into two components:

$$\sigma = (C, S),$$

where Content ($C$) denotes a dataset $D$, a chart type $\tau$, and a mapping function that encodes variables in $D$ to geometric marks. Style ($S$) denotes the visual configuration, including palettes, fonts, strokes/fills, gridlines, legend layout, spacing, and margins.

An atomic edit $e \in \mathcal{E}$ is a total function $f_e : \Sigma \to \Sigma$, with pre-/post-conditions on $(C, S)$. Given an initial specification $\sigma$ with rendered image $I = R(\sigma)$ and an instruction $u$, a model $M$ produces either an image $\widehat{I} = M(I, u)$ or a specification $\widehat{\sigma} = M(I, u)$.

### 3.2 Tasks

**Task 1: Single Chart Edit.** Given $(I, u)$ where $u$ specifies one atomic edit $e$, the updated specification is as follows:

$$\sigma^\star = f_e(\sigma), \qquad I^\star = R(\sigma^\star).$$

**Task 2: Multiple Chart Edits.** Given $(I, u)$ where $u$ specifies $k \geq 2$ atomic edits $\{e_1, \ldots, e_k\}$ applied jointly, the updated specification is

$$\sigma^\star = (f_{e_k} \circ \cdots \circ f_{e_1})(\sigma), \qquad I^\star = R(\sigma^\star).$$

For non-commutative edits, we adopt a fixed canonical order in the generator.

**Task 3: Conversational Chart Edits.** A session consists of $T$ rounds. At round $t$, the input is $(I_{t-1}, H_{t-1}, u_t)$, where $I_{t-1}$ is the previous image, $H_{t-1}$ is the dialogue history, and $u_t$ is the current instruction. The updated specification is

$$\sigma_t^\star = (f_{e_t} \circ \cdots \circ f_{e_1})(\sigma), \qquad I_t^\star = R(\sigma_t^\star).$$

**Task 4: Style Transfer.** Given a source chart $I_s = R(\sigma_s)$ and target content $(D_t, \tau_t)$, the goal is to preserve the target content while adopting the source's style:

$$C(\sigma^\star) = (D_t, \tau_t), \qquad S(\sigma^\star) \approx S(\sigma_s), \qquad I^\star = R(\sigma^\star).$$

**Task 5: Visual-Guidance Edits.** Given $(I, u, \mathcal{G})$, where $\mathcal{G}$ is visual guidance, the goal is to apply the edit $u$ within the guided region while preserving other regions:

$$\sigma^\star = f_{e,u,\mathcal{G}}(\sigma), \qquad I^\star = R(\sigma^\star).$$

### 3.3 Base Figure Sourcing and Generation

To construct base figures, we define a set of chart classes $\mathcal{C}$ and associate them with curated datasets $\mathcal{A}$ drawn from public sources (full list in Appx. F). Each chart class $c \in \mathcal{C}$ is paired with a preference list $\mathcal{P}(c)$ to encourage semantically coherent choices. We employ a LLM to propose candidate specifications conditioned on class hints and dataset lists. A set of automatic validation and filtering rules ensures that generated charts satisfy schema requirements, avoid duplicates, and maintain semantic diversity. In addition, heuristic alignment between dataset domains and chart types further improves quality and coverage. All generations are logged with provenance information, and further implementation details are provided in Appx. A.

### 3.4 Editing Operations

We build a suite of editing tasks derived from a canonical operation set $\mathcal{O}$ (See Appx. B for more details). Each element in $\mathcal{O}$ encodes an atomic edit, covering text, style, layout, and data–centric manipulations. Invalid operations are filtered out depending on chart semantics (e.g., spacing edits require band/point scales).

From each chart we automatically produce (i) natural–language instructions augmented with machine–readable OP tags[3], (ii) edited specifications with inline data values, and (iii) corresponding

---

[3]OP = operation; each OP tag encodes the intended atomic edit.

Table 3: Distribution of editing operations by task. Operations are grouped into categories such as data-centric, text, style, and layout, with counts reported per task and overall totals.

| Task | Category | Operation | Image Count |
|---|---|---|---|
| **Single Edit** | Data-centric | Add element | 1941 |
| | | Remove element | 1892 |
| | Text | Add title | 1942 |
| | Style Editing | Change background color | 1944 |
| | | Change data color | 1729 |
| | Margin Adjustments | Adjust category spacing | 1729 |
| | Font | Font Adjustment | 2943 |
| **Multi Edit** | Dual-operation | Combine 2 edits | 3370 |
| | Triple-operation | Combine 3+ edits | 2660 |
| **Conversational Edit** | | | 3575 |
| **Visual Guidance** | Style Editing | Change data color | 1666 |
| | Data-centric | Remove element | 1819 |
| **Style Transfer** | Style Mapping | Transfer style | 1511 |
| | Style Editing | Change data color | 1728 |
| | Margin Adjustments | Adjust category spacing | 387 |
| | | **Overall Total** | **30836** |

rendered images. On top of these atomic edits, we derive (iv) conversational annotations that align multi–step edits with their constituent single edits, (v) visual–guidance assets where the target region is circled on the original chart, and (vi) style–transfer annotations that pair a target edit with a reference figure providing the desired style attribute. More details are provided in Appx. B.

### 3.4.1 SINGLE AND MULTI EDIT GENERATION

For each chart we sample a feasible subset $\mathcal{O}(c) \subseteq \mathcal{O}$ and realize the edits as natural instructions with corresponding OP tags. Edited specifications are validated to preserve schema correctness, ensure visible changes, and maintain consistent data accounting when adding or removing rows. These checks guarantee deterministic and reproducible supervision.

### 3.4.2 CONVERSATIONAL ANNOTATIONS

We further construct short multi–turn conversations by decomposing a two–step edit into its constituent single edits. Each conversational sample provides the original chart, two turns of instructions with their intermediate ground truth states, and the final outcome. This setting evaluates whether models can maintain state and history across turns rather than only executing isolated edits.

### 3.4.3 VISUAL–GUIDANCE ASSETS

For a selected subset of operations, we create visually grounded variants by marking the target region directly on the original chart. To generate the visual overlay, we employ a vision–language model (GPT-Image) that is prompted to draw a thin red circle around the specified element while leaving chart content unchanged. Each sample provides both a concise natural instruction and a guidance image with the circled target. This variant enables evaluation of multimodal understanding, where the model must integrate textual instructions with explicit visual cues.

### 3.4.4 STYLE–TRANSFER ANNOTATIONS

Finally, we introduce a style–transfer setting in which an edited chart is paired with a reference chart whose current style attribute matches the target of the edit. The model is asked to reproduce the target chart while adopting the style of the reference. This task connects editing with cross–figure style adaptation and highlights the challenge of disentangling content from stylistic attributes.

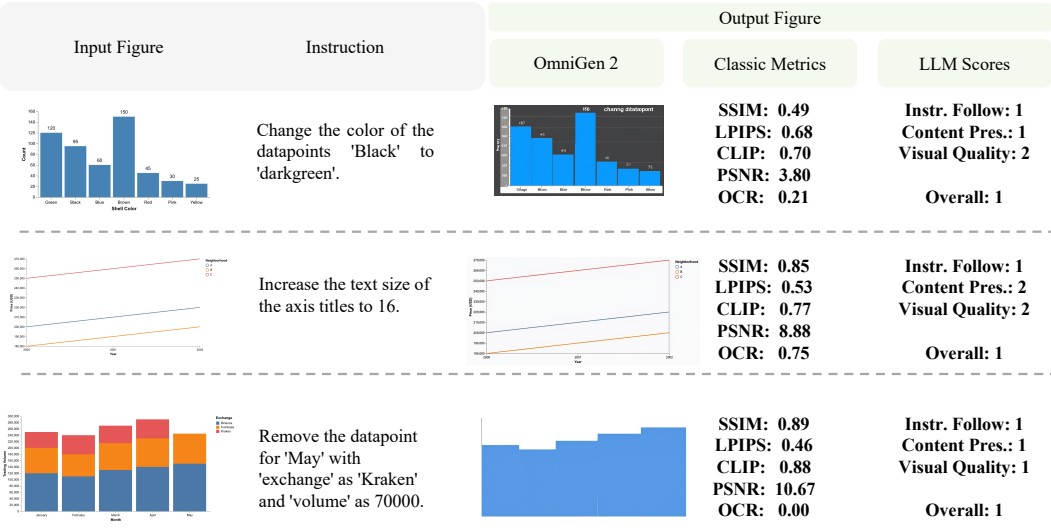

Figure 2: Comparison of chart editing evaluation signals on three representative cases. The left block shows the *Input Figure* and the *Instruction*. The right block shows the *Output Figure* from OmniGen2, the *Classic Metrics* (e.g., SSIM and PSNR), and the *LLM Scores*. We observe that classic pixel metrics can remain high while the edit is wrong. This reveals a gap between pixel similarity and semantic edit correctness, which motivates semantics-aware evaluation for figure editing.

## 3.5 DATASET STATISTICS

The final benchmark contains 30,836 edited figures, distributed across five task families. Tab. 3 summarizes the counts by operation type. Single edits form the largest portion of the dataset, covering basic manipulations such as element addition/removal, text and font changes, color and background modifications, and spacing adjustments, totaling 14,105 figures. Multi edits contribute another 6,244 examples, split between dual edits and three–operation combinations. Conversational settings add 3,732 two–turn sequences, while the visual–guidance and style–transfer tasks contribute 3,355 and 3,400 figures, respectively. Together, these distributions provide a balanced coverage of atomic edits, composite edits, multimodal guidance, and cross–style adaptation. A breakdown by chart type is shown in Tab. 2. Importantly, all base figures are derived from *real-world datasets*, spanning domains such as economics, climate, healthcare, sports, and social science. A complete list of datasets used in figure generation is provided in Appx. F.

## 3.6 EVALUATION PROTOCOL

We evaluate all models directly in image space. We compute six complementary metrics: SSIM (Wang et al., 2004), PSNR (Hore & Ziou, 2010), LPIPS (Zhang et al., 2018), CLIP similarity (Radford et al., 2021), OCR similarity (Smith, 2007), and an LLM-based instruction score. The first five are classic metrics widely used in image generation and vision tasks, while the last directly evaluates whether edits satisfy the instruction, preserve chart content, and maintain visual quality. More details on implementations are provided in Appx. C.

## 4 EXPERIMENT

**Baselines.** We evaluate against four representative instruction-based editing models: GPT-Image (OpenAI, 2025), Imagen 4 (Google, 2025), OmniGen 2 (Wu et al., 2025), and Instruct-Pix2Pix (Brooks et al., 2023b). These span closed–source commercial systems and open–source research frameworks, covering both diffusion-based editors and multimodal approaches. Further details on each baseline are provided in Appx. D.

**Experiment Setup.** We evaluate chart editing across five tasks. All methods operate on the same set of instructions and images. Prompts are standardized to encourage strictly local modifications

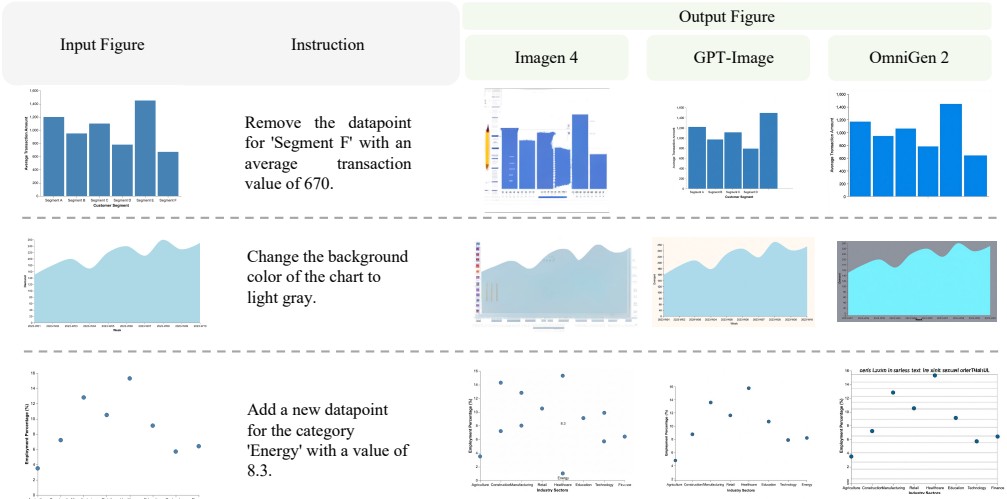

Figure 3: Qualitative examples of figure editing with three representative instructions. For each case, the input figure and target instruction are shown on the left, and outputs from Imagen 4, GPT-Image, and OmniGen2 are shown on the right.

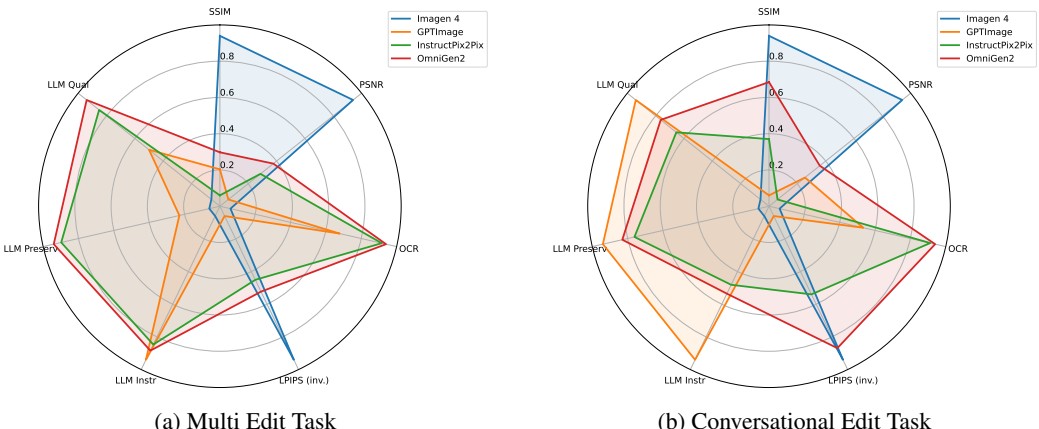

| (a) Multi Edit Task | (b) Conversational Edit Task |
| --- | --- |

Figure 4: Radar charts for different tasks (normalized with epsilon, LPIPS inverted). Each chart compares all models on SSIM, PSNR, OCR, LPIPS, and three LLM scores.

while maintaining axes, labels, and other contextual elements. Further implementation details are provided in Appx. D.

## 4.1 MAIN RESULTS

**Overall performance across tasks.** Tab. 4 summarizes the performance of representative editing models across the five evaluation settings. Imagen 4 achieves consistently high scores on SSIM and PSNR, reflecting strong pixel-level resemblance to the input figures, but its instruction-following and preservation scores are the lowest among all models. GPT-Image excels in conversational and transfer settings, showing the highest instruction-following scores, but often sacrifices content fidelity. OmniGen2 strikes a balance, performing reliably across most tasks with solid LLM scores and relatively stable OCR accuracy. InstructPix2Pix remains competitive but generally underperforms OmniGen2, particularly on complex edits, while still clearly surpassing Imagen 4 on semantic alignment. These results highlight that strong performance on pixel-based similarity metrics does not necessarily translate into correct or faithful edits.

**Limitations of classic metrics.** Fig. 4 provides a more detailed comparison of multi-edit and conversational tasks. Classic metrics such as SSIM and PSNR exaggerate the performance of pixel-oriented models like Imagen 4, while LLM-based scores and OCR accuracy reveal significant se-

Table 4: Performance comparison grouped by task. Higher is better for SSIM, CLIP, PSNR, OCR, and LLM Scores. Lower is better for LPIPS. **Instr.** denotes instruction following score. **Preserv.** denotes content preservation score. **Qual.** denotes image quality score.

| Task | Model | SSIM ↑ | LPIPS ↓ | CLIP ↑ | PSNR ↑ | OCR ↑ | MLLM Score (1–5) ↑ | | |
|------|-------|--------|---------|--------|--------|-------|-------|----------|-------|
| | | | | | | | Instr. | Preserv. | Qual. |
| **Single** | Imagen 4 | **0.7726** | **0.4094** | 0.7781 | **13.04** | 0.0723 | 1.58 | 1.51 | 2.05 |
| | GPTImage | 0.7295 | 0.5383 | 0.8099 | 10.32 | 0.2054 | **3.47** | 1.71 | 2.45 |
| | InstructPix2Pix | 0.7211 | 0.4811 | 0.8328 | 11.02 | 0.2568 | 3.27 | 2.50 | 2.77 |
| | OmniGen2 | 0.7350 | 0.4705 | **0.8350** | 11.30 | **0.2620** | 3.35 | **2.55** | **2.85** |
| **Multi** | Imagen 4 | 0.6958 | 0.5549 | 0.7738 | **11.02** | 0.1069 | 1.26 | 1.32 | 2.15 |
| | GPTImage | 0.7017 | 0.5787 | **0.8070** | 9.73 | 0.2185 | 2.51 | 1.63 | 2.34 |
| | InstructPix2Pix | 0.6460 | 0.5204 | 0.8043 | 9.83 | 0.2584 | 2.48 | 2.00 | 2.51 |
| | OmniGen2 | **0.7100** | **0.5100** | 0.8220 | 10.15 | **0.2650** | **2.65** | **2.10** | **2.70** |
| **Conv.** | Imagen 4 | **0.7180** | **0.4923** | 0.7599 | **11.58** | 0.0698 | 1.35 | 1.23 | 2.11 |
| | GPTImage | 0.6732 | 0.5257 | **0.8525** | 10.66 | 0.1721 | **4.59** | **2.51** | **2.91** |
| | InstructPix2Pix | 0.6890 | 0.5075 | 0.8200 | 10.40 | 0.2540 | 2.90 | 2.25 | 2.65 |
| | OmniGen2 | 0.7050 | 0.4950 | 0.8280 | 10.80 | **0.2600** | 3.10 | 2.35 | 2.75 |
| **Visual** | Imagen 4 | **0.8420** | 0.5050 | 0.7600 | **13.10** | 0.1200 | 1.40 | 1.35 | 2.20 |
| | GPTImage | 0.8355 | 0.5207 | **0.8444** | 12.85 | **0.4665** | **2.39** | **3.16** | **3.95** |
| | InstructPix2Pix | 0.7380 | 0.5220 | 0.8190 | 10.90 | 0.2200 | 1.85 | 2.20 | 2.80 |
| | OmniGen2 | 0.7508 | 0.5236 | 0.8187 | 8.98 | 0.1806 | 1.19 | 1.85 | 2.74 |
| **Transfer** | Imagen 4 | **0.8500** | 0.4800 | 0.7700 | **14.00** | 0.1300 | 1.30 | 1.25 | 2.15 |
| | GPTImage | 0.8438 | 0.4934 | 0.8054 | 13.81 | **0.5092** | **3.06** | **3.57** | **4.16** |
| | InstructPix2Pix | 0.7960 | 0.5020 | **0.8160** | 12.90 | 0.2400 | 2.20 | 2.60 | 3.10 |
| | OmniGen2 | 0.8246 | **0.4376** | 0.8127 | 12.08 | 0.3147 | 1.53 | 2.14 | 2.64 |

mantic errors. The radar plots make this gap visually explicit: models that appear strong under pixel similarity collapse when judged by whether the requested edits were actually applied. This finding is consistent with the qualitative evidence in Fig. 3 and reinforces the need for evaluation protocols that go beyond pixel resemblance.

**Per-instruction breakdown.** We further analyze performance at the level of individual instructions in Appx. E. These results confirm the same trend: models often achieve high SSIM or PSNR even when edits such as adding datapoints or changing axis labels are not correctly applied.

## 4.2 ANALYSIS

**The gap between pixel-level similarity and semantic correctness.** Fig. 2 and Tab. 4 highlight a consistent limitation of classic image metrics in the context of figure editing. Models such as Imagen 4 and OmniGen2 can obtain high SSIM and PSNR scores, yet their outputs often fail to apply the intended transformation. As illustrated in Fig. 2, edits may preserve overall appearance while the instruction is ignored, the figure is distorted, or key content is changed. Tab. 4 shows the same pattern across tasks: pixel-based metrics remain strong, but instruction-following and content-preservation scores from LLM-based evaluation drop sharply, especially for multi-step and conversational edits. These results indicate that similarity at the pixel level is not a reliable indicator of semantic correctness. They also motivate the need for benchmarks that evaluate edits at the level of data and visual encodings rather than image resemblance alone.

**No single model dominates across tasks.** Tab. 4 shows that performance is highly fragmented: no model achieves consistently strong results across all task types or metrics. Imagen 4 tends to lead on low-level pixel fidelity metrics such as SSIM and PSNR, yet it performs poorly on instruction-following and semantic preservation, indicating that its edits often look visually smooth but fail to reflect the requested change. GPT-Image shows the opposite trend: it excels in instruction scores, especially in conversational and transfer settings, but lags behind on PSNR and OCR accuracy, suggesting weaker robustness to text-heavy or layout-sensitive edits. InstructPix2Pix performs competitively on some semantic metrics but is generally less reliable than OmniGen2, which offers a more

balanced profile. However, OmniGen2 also struggles with visual-guided and transfer edits, highlighting its limitations in cross-instance reasoning. These results reveal that current models overfit to specific task structures or metric types, and that strong performance on classic pixel-level metrics does not guarantee reliable edit satisfaction in more challenging scenarios.

**Qualitative study.** Fig. 3 illustrates representative failure cases in figure editing. Across different instructions: removing a datapoint, changing a background color, or adding a new element, current models frequently produce outputs that appear visually similar yet fail to realize the requested transformation. These cases mirror the quantitative results: classic pixel-level metrics often remain high even when semantic correctness is violated. The examples highlight how generative editors, optimized for perceptual similarity, struggle with structure-preserving transformations, reinforcing the need for evaluation protocols and benchmarks that explicitly target semantic consistency. More cases can be found in Appx. 5.

## 5    CONCLUSION

We introduced *FigEdit*, a large-scale benchmark for scientific figure editing that treats editing as a structured transformation problem grounded in graphical grammar. The benchmark spans diverse chart types and task settings, and it provides both figure outputs and executable specifications to support reliable evaluation. Our experiments show that existing models perform poorly when edits require semantic consistency, which reveals a clear gap between current approaches and the needs of figure editing. By offering a task-structured and semantics-aware evaluation protocol, *FigEdit* establishes a foundation for developing future models that can perform faithful, data-aligned, and auditable edits.

## 6    ETHICS STATEMENT & REPRODUCIBILITY STATEMENT

This work adheres to standard academic research practices. All data used are either publicly available or synthetically generated, and the study is intended solely for scientific and educational purposes. We do not foresee any ethical concerns arising from the content or methodology presented. For reproducibility, we have included sufficient technical details in the paper to allow other researchers to replicate our experiments. The dataset statistics, task definitions, and evaluation protocols are described in detail, and we aim to facilitate further exploration and extension by the community.

## 7    ACKNOWLEDGEMENT

We sincerely appreciate the support from Adobe Research and the USC FORTIS Lab.

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

## A  BASE FIGURE SOURCING AND GENERATION

As discussed in Sec. 3.3, base figures are generated for chart classes $\mathcal{C}$ (bar, stacked–bar, line, area, box, violin, donut, pie, dot, scatter) using dataset names from a curated whitelist $\mathcal{A}$ (see Appx. F). For each class $c \in \mathcal{C}$, a preference list $\mathcal{P}(c) \subseteq \mathcal{A}$ guides the assignment toward semantically coherent themes.

**LLM–guided spec proposal.**  A chat model $M$ is instructed to output a single JSON object

$$o = \{\texttt{vega\_spec} = \sigma, \ \texttt{dataset} = d\}, \qquad \sigma \in \Sigma, \ d \in \mathcal{A},$$

where $\mathcal{A}$ is the set of allowed dataset names. Any mismatch with the requested dataset $d$ triggers rejection and re-sampling. Each prompt includes a class hint $H(c)$, a preferred dataset list $\mathcal{P}(c)$, an exemplar specification $E_c$ (style only), and an *avoid–terms* block derived from recent generations. The sampling temperature is fixed to $\tau = 0.55$ to balance validity and diversity. Detailed prompt template is shown below:

---

### System Message (Part 1)

```
1  You return ONLY a valid JSON object with exactly two fields:
2    - vega_spec: a VALID Vega v6 specification JSON object
3    - dataset: ONE string chosen ONLY from the allowed set (see
         Appendix A, Table X)
4
5  Return exactly one JSON object. Do not add any prefix, suffix, or
         code fences.
6  No Markdown, no explanations, no backticks.
7
8  vega_spec requirements:
9    - Use "$schema": "https://vega.github.io/schema/vega/v6.json".
10   - Base data on a popular public dataset; URLs are disallowed.
         Include a small
11     inline sample in "values".
12   - The chosen dataset MUST naturally support category->value or
13     category x series->value aggregation suitable for bar/stacked/
         grouped charts.
14   - Match the requested chart class:
15       * bar: one categorical field + one quantitative field
16       * stacked-bar: category + series + value (stacked)
17       * grouped-bar: category + series + value (side-by-side)
18   - Include all necessary components (data, scales, axes, marks) so
         it renders.
19   - Do NOT include extra meta fields inside vega_spec.
```

**System Message (Part 2)**

```
1  Self-check before responding:
2   - Prefer a specific dataset name from the allowed set that
        credibly aligns
3     with field/entity names (full whitelist in Appendix A, Table X,
          see \ref{appx:allowed-datasets}).
4   - Use "unknown" only if no credible alignment exists.
5
6  Output rules:
7   - Return EXACTLY one JSON object containing { "vega_spec": ..., "
        dataset": ... }.
8   - vega_spec must be a valid Vega v6 JSON object with inline "
        values" (no "url").
9   - No Markdown, no commentary, no extra keys.
```

**User Template (Bar, excerpt)**

```
1  Task:
2  - Chart class: {chart_class}
3  - Instruction: {class_hint}
4  - Generate a JSON object with two fields: vega_spec (Vega v6 spec)
        and dataset
5    (string from the allowed set or 'unknown').
6
7  Hard constraint for this sample:
8  - You MUST set "dataset" EXACTLY to: {dataset_target}
9  - Field names and inline 'values' must be plausible for {
        dataset_target}.
10
11 Data requirements:
12 - Include inline 'values' only (no 'url'); fit bar => category +
        value.
13 - Use 5..12 categories; numeric magnitudes should be plausible for
        the dataset.
14
15 Diversity controls:
16 - Avoid reusing identical numeric multisets for the same field set.
17 - Avoid terms seen recently:
18 {avoid_terms}
19
20 Preferred datasets for this class (see Appendix A, Table X for the
        full list):
21 {preferred_datasets}
22
23 Output:
24 - ONLY one JSON object: { "vega_spec": ..., "dataset": ... }.
```

**Scheduling and validity.** A scheduler balances dataset usage by always selecting the least-used candidate for each chart class, based on compatibility heuristics (e.g., time series $\mapsto$ line/area; survey data $\mapsto$ bar/pie/dot). Returned specifications are checked for Vega v6 schema, completeness (`data`, `marks`, `scales`, `axes`), and type-specific field patterns (e.g., bar requires {category, numeric}; stacked–bar requires {category, series, numeric}). Invalid proposals are rejected and resampled.

**Shape validation.** Beyond generic schema checks, additional constraints enforce meaningful content. For example, bar charts must contain at least one categorical and one numeric field, while

stacked–bar charts must include two categorical fields and one numeric field. Other chart types are validated using generic rules.

**Duplicate and near–duplicate control.**   For every $\sigma$, we compute four signatures over its inline data and structure:

$$h_{\text{exact}}(\sigma) = \text{SHA256}\Big(\bigoplus_{\text{rows}} \text{ sorted Vega rows}\Big),$$
$$h_{\text{multi}}(\sigma) = \text{SHA256}(\text{numeric multiset per field set}),$$
$$s_{\text{val}}(\sigma) = \text{SHA256}(\text{per–field histograms with } b{=}6 \text{ and } (\mu, \sigma)),$$
$$s_{\text{struct}}(\sigma) = \text{SHA256}(\text{size buckets, mark types, scale types/flags, axis orients, legend presence}),$$

where SHA256 is a cryptographic hash function that produces a fixed 256-bit digest with extremely low collision probability. A specification is rejected if $h_{\text{exact}}$ or $h_{\text{multi}}$ has been observed previously, or if both $s_{\text{val}}$ and $s_{\text{struct}}$ have appeared before. This eliminates duplicates and near–duplicates while permitting controlled variability.

**Semantic diversity via term overlap.**   Categorical fields are inferred from scales and encodings, forming a token set $T(\sigma)$. A sliding window $\mathsf{W}$ of the last $k$ samples (default $k = 16$) is maintained, and the Jaccard overlap ratio

$$r = \frac{|T(\sigma) \cap U|}{|T(\sigma) \cup U|}, \qquad U = \bigcup_{S \in \mathsf{W}} S$$

is computed. A candidate is rejected if $r > \theta$ (default $\theta = 0.70$) and $|T(\sigma) \setminus U| < m$ (default $m = 2$). The current union $U$ is injected back into subsequent prompts as an *avoid–terms* block, enforcing semantic diversity across generations.

**Additional mechanisms.**   Further enhancements improve robustness: malformed completions are handled by stripping Markdown fences or extracting JSON blocks; provenance is logged into a JSON index with raw outputs for debugging. Together, these mechanisms ensure quality, diversity, and reproducibility of the generated base figures.

## B   EDITING OPERATIONS

As briefly discussed in Sec. 3.4, this section provides extended details of the editing operations and annotation pipeline. We describe how we generated single and multi-edit supervision, conversational annotations, visual-guidance assets, and style-transfer pairs. Representative prompt excerpts are also included. We first define a canonical operation set:

$$\mathcal{O} = \left\{ \begin{array}{ll} \texttt{change\_datapoint\_color,} & \texttt{increase\_text\_size,} \\ \texttt{decrease\_text\_size,} & \texttt{change\_background\_color,} \\ \texttt{increase\_category\_spacing,} & \texttt{decrease\_category\_spacing,} \\ \texttt{add\_title,} & \texttt{add\_datapoint,} \\ \texttt{remove\_datapoint} & \end{array} \right\}.$$

### B.1   SINGLE & MULTI EDIT GENERATION

For each chart specification we select a feasible subset $\mathcal{O}(c)$, filtering out inapplicable edits (e.g., spacing operations for charts without band/point scales, or removals when only one data row exists). An LLM is prompted to return exactly one sentence instruction followed by explicit OP tags, as well as the edited Vega v6 specification. We canonicalize op names, infer missing keys (such as `axis_label_size`, `new_color`, `new_bg`, or `new_padding`), and apply minimal but deterministic edits to ensure the modification is visually effective. Validation includes schema conformance, key completeness, and visible effect realization. Detailed prompt is shown below:

**Prompt (Single/Multi, Part 1)**

```
1   Return ONLY a JSON object (no markdown) with keys:
2   - "instruction": ONE English sentence that describes exactly N
        edits (N in {1,2,3}),
3     followed by EXACTLY N tag lines in order:
4       [#OP1 op=<...>; key1=value1; key2=value2; ...]
5       [#OP2 op=<...>; ...]
6       [#OP3 op=<...>; ...] (only if N==3)
7   - "ops": array of length N; each item has "op" in:
8     ["change_datapoint_color","increase_text_size","decrease_text_size
        ",
9      "change_background_color","increase_category_spacing","
          decrease_category_spacing",
10     "add_title","add_datapoint","remove_datapoint"]
11  - "edited_spec": a VALID Vega v6 JSON spec that keeps "$schema" v6
        and uses inline "values" only
12    (strictly no "url").
13
14  Required keys per operation (inside each [#OPi ...] tag):
15  - change_background_color:
16      new_bg=<css-or-#hex>
17  - increase_text_size / decrease_text_size:
18      axis_label_size=<int in 6..30>; tick_size=<int> (optional);
          title_size=<int> (optional)
```

---

**Prompt (Single/Multi, Part 2)**

```
1   - increase_category_spacing / decrease_category_spacing:
2       scale=auto|x|y; new_padding=<float in [0,0.9]>
3   - add_title:
4       title_text=<non-empty>; title_size=<int> (optional)
5   - change_datapoint_color:
6       target_category=<label>; target_series=<label> (omit if not
            applicable);
7       new_color=<css-or-#hex not used in original>
8   - add_datapoint:
9       If single-series:
10        position=before:<existing>|after:<existing>|end;
11        category=<new_label>; value=<number>
12      If multi-series:
13        position=...; category=<cat_label>; series=<series_label>;
              value=<number>
14  - remove_datapoint:
15      If single-series:
16        category=<existing_label>
17      If multi-series:
18        category=<existing_label>; series=<existing_series>
19
20  Editing rules and checks:
21  - Apply minimal, deterministic edits; ensure each step has a
        visible effect.
22  - Maintain inline data only; never introduce "url".
23  - Sizes must be >= 6; band/point padding must be within [0,0.9].
24  - New colors must not collide with original literal colors.
25  - For add/remove datapoint, edit exactly one row and keep row-count
        accounting consistent:
26      rows(edited) = rows(original) + adds - removes
27  - Preserve unrelated content (data, labels, titles) unless the step
        explicitly changes them.
28
29  For Single set N=1; for Multi set N in {2,3}. Output exactly one
        JSON object.
```

## B.2 CONVERSATIONAL ANNOTATIONS

To simulate multi-turn editing, we align each two-op edit with its corresponding single-op edits. Given a two-op edit $(o_1, o_2)$, we locate the two single edits with the same operations, generate intermediate ground truth images, and concatenate them into a two-round dialogue. Each conversational sample therefore contains: (i) the original figure, (ii) turn-1 with an instruction and intermediate ground truth, and (iii) turn-2 with a follow-up instruction and the final ground truth. This design yields per-round supervision and enables evaluation of temporal consistency. Detailed prompt is shown below:

Prompt (Conversation)

```
You are a strict formatter that assembles a 2-turn conversation
    object
from provided edit annotations. You MUST return ONLY one JSON
    object.

Inputs (conceptual):
- original: the unedited chart (spec+image).
- single-edits: two entries, each has:
    op (atomic op name),
    instruction (one sentence + [#OP* ...] tags),
    edited_spec (Vega v6, inline values only),
    edited_image.
- multi-edit (2-step): has:
    ops=[op1, op2] in the exact execution order,
    edited_spec (final target),
    edited_image,
    instruction (one sentence + tags).

Formatting rules:
1) Preserve execution order strictly as [op1, op2].
2) Use the single-edits whose op matches op1 and op2, respectively.
3) For each turn i in {0,1}:
    - instruction: copy the corresponding single's instruction
        verbatim,
     trimming leading/trailing whitespace only.
    - gt: spec=image=the corresponding single's ground truth (
        intermediate).
    - op: the corresponding op (op1 for turn 0, op2 for turn 1).
4) final: use the multi-edit's edited_spec and edited_image.
5) Include the multi-edit's instruction as multi_instruction.
6) Do NOT invent or rewrite text; do NOT change specs.
7) Output must be a single JSON object with the following fields:

{
  "chart_type": "<string>",
  "figure_id": "<string>",
  "ops": ["<op1>", "<op2>"],
  "original": {"spec": <json>, "image": "<path-or-id>"},
  "turns": [
    {
      "turn_idx": 0,
      "op": "<op1>",
      "instruction": "<single1_instruction_trimmed>",
      "gt": {"spec": <json>, "image": "<path-or-id>"}
    },
    {
      "turn_idx": 1,
      "op": "<op2>",
      "instruction": "<single2_instruction_trimmed>",
      "gt": {"spec": <json>, "image": "<path-or-id>"}
    }
  ],
  "final": {"spec": <json>, "image": "<path-or-id>"},
  "multi_instruction": "<multi_instruction_trimmed>"
}

Return exactly this one JSON object and nothing else.
```

B.3    VISUAL–GUIDANCE ASSETS

For selected atomic operations (notably datapoint color changes and datapoint removals), we construct visual-guided variants by highlighting the target region directly in the original chart. To produce the overlays, we employ a vision–language model (GPT-Image) instructed to draw a thin red circle around the specified element while leaving the rest of the chart untouched. This yields paired data: (i) a natural-language instruction referencing the circled element, and (ii) a visually annotated chart. Such assets allow evaluation of multimodal understanding, where the model must integrate textual instructions with explicit visual cues. Detailed prompt is shown below:

**Prompt (Visual Guidance)**

```
1  You are an image editor. Given a chart image and a target
       description,
2  draw a thin red circle around exactly one target element. Do not
       change
3  any chart content.
4
5  Inputs:
6  - Image: the original chart may be letterboxed on a plain
       background.
7  - Chart noun: {bar|slice|point|mark}.
8  - Target condition (optional but preferred):
9    category == "<CATEGORY>"
10    series == "<SERIES>"
11
12 Task:
13 - Locate the single element that satisfies the target condition (if
        given).
14 - If no explicit condition is given, use the instruction sentence
       prefix
15   as a hint to identify the most likely target element.
16 - Draw exactly one circle that tightly encloses the target element.
17
18 Rendering constraints:
19 - Stroke color: #FF0000 (pure red).
20 - Stroke style: thin line, no glow, no shadow.
21 - Circle only; no arrows, no text, no highlights or masks.
22 - Do not crop, scale, or move the chart content.
23 - If the image is letterboxed, ignore padding/margins/borders and
       place
24   the circle over the chart area only.
25 - Preserve the original resolution and aspect ratio.
26 - Do not alter colors, fonts, or data marks other than the added
       circle.
27
28 Output:
29 - Return a single edited image where the only modification is the
       thin
30   red circle tightly around the target element.
```

B.4    STYLE–TRANSFER SINGLES

We further derive one-shot style-transfer supervision by linking existing single edits to style sources. For each single edit, we identify another original chart whose current style attribute already matches the target attribute of the edited chart. We construct a natural instruction such as "Make this bar chart use the same background color as the reference chart," and pair it with the corresponding OP

tag. This produces style-transfer pairs across both same-type and cross-type chart classes, enabling evaluation of style generalization. Detailed prompt is shown below:

---

**Prompt (Style Transfer)**

```
1  You are writing ONE natural-language instruction for a style
       transfer (single attribute).
2  Inputs you are given (out of band) include:
3  - target chart class and image,
4  - a SINGLE-OP edit already chosen for the target (with its OP tag
       line),
5  - a reference chart ("the reference chart") that already exhibits
       the target style value.
6
7  Task:
8  - Write exactly ONE concise English sentence that asks to make the
       target chart
9    use the SAME style attribute as the reference chart.
10 - Use phrasing like:
11     "Make this <chart noun> use the same background color as the
           reference chart."
12     "Make this <chart noun> use the same axis label font size as the
            reference chart."
13     "Make this <chart noun> use the same category spacing as the
           reference chart."
14     "Match the datapoint color used in the reference chart."
15     "Add a title with the same font size as the reference chart."
16 - Do NOT mention internal ids. Say "the reference chart" or "the
       example chart".
17
18 After the sentence, append EXACTLY ONE OP tag line, keeping the
       provided tag VERBATIM:
19   [#OP1 op=<one_of_allowed_ops>; key1=value1; key2=value2; ...]
20
21 Allowed ops (single attribute only):
22   change_background_color | increase_text_size | decrease_text_size
         |
23   increase_category_spacing | decrease_category_spacing |
24   add_title | change_datapoint_color
25
26 Hard constraints:
27 - Output ONLY the final instruction text (one sentence) followed by
       the single OP tag line.
28 - Do NOT return JSON. Do NOT include edited_spec. Do NOT invent new
       keys or values.
29 - Preserve the original OP tag exactly as given (verbatim).
```

---

Through these pipelines, each figure can appear as (i) atomic edits (single/multi), (ii) conversational trajectories, (iii) visually guided variants, and (iv) style-transfer pairs. All assets are designed to be reproducible, diverse, and machine-readable, while supporting multimodal evaluation settings.

## C  EVALUATION METRICS

As discussed in Sec. 3.6, we report both classic image metrics and an LLM-based score to capture semantic correctness.

**SSIM.** Structural Similarity Index Wang et al. (2004) is applied on grayscale renderings with Gaussian weighting to emphasize local structure. This metric accounts for luminance, contrast, and structure, making it more perceptually meaningful than raw pixel errors.

**PSNR.** Peak Signal–to–Noise Ratio Hore & Ziou (2010) is computed with pixel values clipped to $[0, 255]$ and averaged across RGB channels. It quantifies the logarithmic ratio between the maximum possible signal and mean squared error.

**LPIPS.** Learned Perceptual Image Patch Similarity Zhang et al. (2018) is computed using the official framework, with AlexNet as the default backbone. Images are normalized to $[-1, 1]$ before feature extraction. LPIPS captures perceptual discrepancies such as texture or shape distortions.

**CLIP similarity.** We use CLIP ViT-L/14 Radford et al. (2021) to extract image embeddings and report cosine similarity between $\widehat{I}$ and $I^\star$. This provides a semantic-level measure of alignment beyond pixel similarity.

**OCR similarity.** We extract text from both images using Tesseract OCR Smith (2007). Similarity is measured as the normalized edit distance:

$$\text{Sim}_{\text{OCR}} = 1 - \frac{\text{EditDist}(s_{\widehat{I}}, s_{I^\star})}{\max(|s_{\widehat{I}}|, |s_{I^\star}|)},$$

where $s_{\widehat{I}}$ and $s_{I^\star}$ are the concatenated OCR strings. This metric emphasizes correctness of labels, legends, and annotations.

**LLM-based instruction score.** To directly evaluate editing success, we prompt a large language model OpenAI (2024) with (i) the original chart and instruction $(I, u)$, (ii) the edited output $\widehat{I}$, and (iii) the ground truth $I^\star$. The model issues binary judgments on:

- Instruction satisfaction: whether the requested edit is applied.

- Content preservation: whether the underlying chart data remain intact.

- Visual quality: whether the rendering is artifact-free and coherent.

Responses are parsed into structured JSON objects, which are aggregated into per-instance and per-model scores. Trimmed prompt examples are provided below:

---

**LLM Score (System Message, Part 1)**

```
1  You are an expert AI assistant specializing in data visualization
      evaluation.
2  Your task is to evaluate how well an AI-generated chart follows a
      given text instruction.
3  You will be given an instruction, a reference "Ground Truth" image,
      and the "Generated Image" to evaluate.
4
5  Evaluate the generated image based on the following three criteria:
6
7  1. Instruction Following (score_instruction): How accurately was
      the specific
8    instruction executed? (e.g., if asked to change color to orange,
        is it orange?)
9  2. Content Preservation (score_preservation): Were all other
      elements of the
10   chart preserved correctly without unwanted changes? (e.g., data
        values,
11   labels, and titles are unchanged unless specified).
12 3. Image Quality (score_quality): Is the generated image free of
      major artifacts,
13   distortions, or unreadable text?
```

---

---

**LLM Score (System Message, Part 2)**

```
1  For each of the above, assign a score from 1 (very poor) to 5 (
       excellent).
2  Then compute a total score (score) as the average of the three
       above,
3  rounded to the nearest integer.
4
5  Your response MUST be a JSON object with the following keys:
6  - "score_instruction": Integer [1-5]
7  - "score_preservation": Integer [1-5]
8  - "score_quality": Integer [1-5]
9  - "score": Integer [1-5], the average of the above
10 - "reasoning": One-sentence explanation justifying the scores
11
12 Example Response:
13 {
14     "score_instruction": 5,
15     "score_preservation": 4,
16     "score_quality": 5,
17     "score": 5,
18     "reasoning": "The instruction was followed perfectly, content
           was mostly
19     preserved, and the image quality is excellent."
20 }
```

---

**LLM Score (User Template)**

```
1  **Instruction:**
2  <instruction text>
3
4  **Reference Image (Ground Truth):**
5  <data:image/png;base64,...>
6
7  **Generated Image (to be evaluated):**
8  <data:image/png;base64,...>
```

---

## D  ADDITIONAL EXPERIMENTAL DETAILS

**Pre- and Post-Processing.** To ensure consistent inputs, all charts are letterboxed into a square canvas before inference. After editing, outputs are mapped back to the original resolution using contain resizing, which preserves the full layout without cropping. This procedure guarantees that models are evaluated under identical geometric conditions while avoiding distortion of axes or labels.

**Prompt Construction.** For all tasks, prompts explicitly instruct the model to make localized modifications while leaving unrelated elements unchanged. In Visual tasks, prompts additionally emphasize that only the circled region should be modified. For Transfer tasks, the prompt specifies a two-panel setup, where only the left (base) panel is editable and the right (reference) panel serves as a style guide.

**Baselines.** For comparison, we include four representative baselines that capture the current state of instruction-driven image editing: (1) GPT-Image (OpenAI, 2025). A commercial instruction–driven editing system provided by OpenAI. It supports free-form natural language instructions and has been widely used for general-purpose editing tasks. Although proprietary, it reflects the strongest available commercial option. (2) Imagen 4 (Google, 2025). A proprietary diffusion–based editor developed by Google and released via the Vertex AI platform. Imagen 4 is optimized for control-

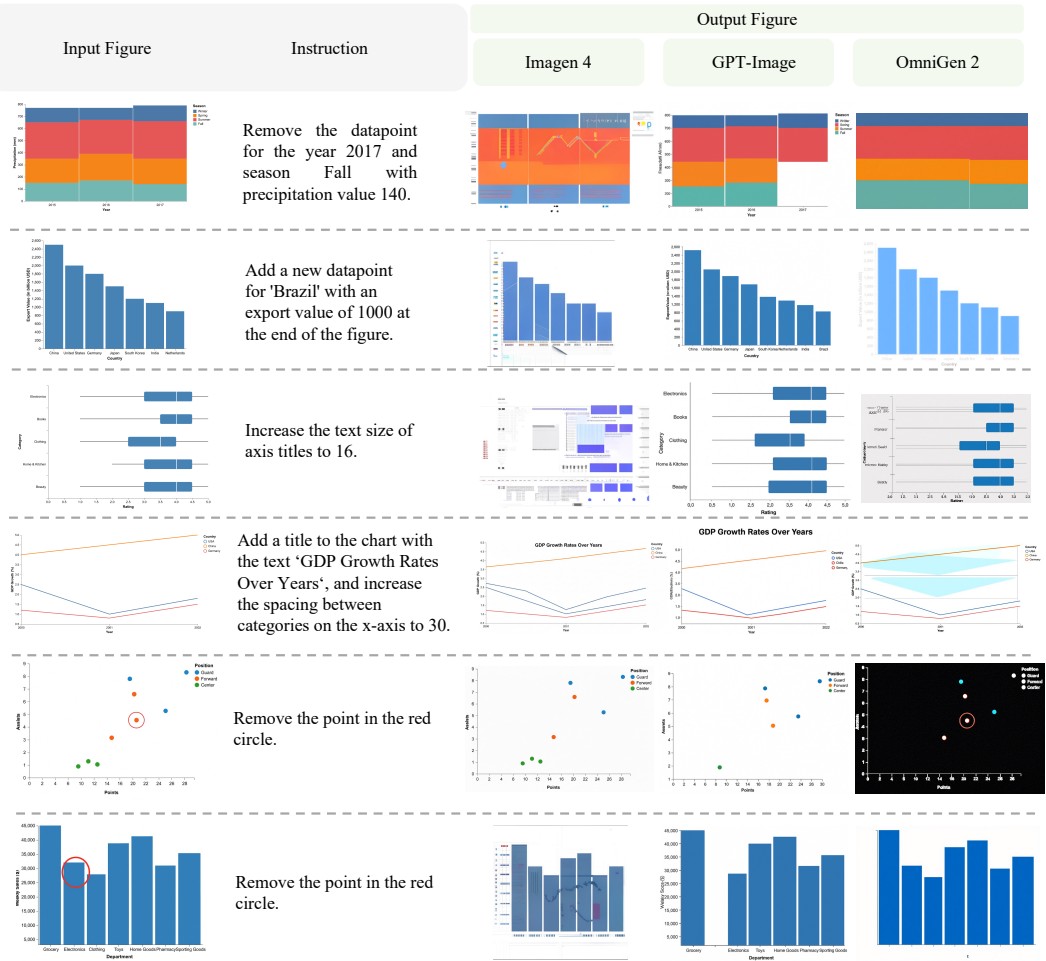

Figure 5: Additional qualitative examples of figure editing results. Each row shows an input figure (left), the corresponding natural language instruction (middle), and the output figures generated by Imagen 4, GPT-Image, and OmniGen 2 (right). The cases cover representative edit types, including data point removal, data point addition, axis text scaling, layout adjustments, and targeted point deletion. While the models sometimes produce visually consistent outputs, they often fail to accurately execute the requested transformation, highlighting the limitations of current instruction-based figure editing systems.

lable, high-fidelity image generation and editing, though its design is primarily tuned for natural image content. (3) OmniGen 2 (Wu et al., 2025). An open–source multimodal model recently introduced for text-guided and image-guided editing. It supports multi-turn interaction and has shown promising results for chart and figure editing. We use the official released checkpoint and inference pipeline. (4) InstructPix2Pix (Brooks et al., 2023b). An open–source approach that finetunes a diffusion backbone on paired instruction–image data. It was among the first methods to explicitly align natural language instructions with image translation, and remains a strong research baseline for instruction-conditioned editing.

Together, these baselines span both closed and open ecosystems, diffusion and multimodal paradigms, and commercial and academic settings. They represent the strongest available instruction-driven editing approaches at the time of writing.

Table 5: Per-instruction performance comparison (Part 1/2). Higher is better for SSIM, CLIP, PSNR, OCR, and LLM Scores. Lower is better for LPIPS. **Instr.** denotes instruction following score. **Preserv.** denotes content preservation score. **Qual.** denotes image quality score.

| Model | SSIM ↑ | LPIPS ↓ | CLIP ↑ | PSNR ↑ | OCR ↑ | LLM Score (1–5) ↑ | | |
| | | | | | | Instr. | Preserv. | Qual. |
|---|---|---|---|---|---|---|---|---|
| **Instruction: Change the colors of the data point** | | | | | | | | |
| InstructPix2Pix | 0.736 | 0.52 | 0.84 | 10.7 | 0.25 | 2.66 | 2.76 | 2.76 |
| OmniGen2 | 0.748 | 0.48 | 0.85 | 11.1 | 0.26 | 3.63 | 3.32 | 3.66 |
| GPT-Image | 0.733 | 0.54 | 0.87 | 10.4 | 0.22 | 4.34 | 3.84 | 4.29 |
| Imagen 4 | 0.772 | 0.41 | 0.80 | 13.1 | 0.08 | 2.09 | 1.84 | 2.72 |
| **Instruction: Add a new title** | | | | | | | | |
| InstructPix2Pix | 0.741 | 0.47 | 0.83 | 11.0 | 0.17 | 1.09 | 2.91 | 3.07 |
| OmniGen2 | 0.744 | 0.46 | 0.84 | 11.2 | 0.29 | 3.34 | 3.14 | 3.36 |
| GPT-Image | 0.728 | 0.53 | 0.88 | 10.5 | 0.36 | 4.91 | 4.43 | 4.41 |
| Imagen 4 | 0.769 | 0.40 | 0.79 | 13.0 | 0.07 | 1.00 | 1.41 | 2.07 |
| **Instruction: Increase font size** | | | | | | | | |
| InstructPix2Pix | 0.735 | 0.50 | 0.84 | 10.8 | 0.26 | 2.48 | 2.10 | 2.83 |
| OmniGen2 | 0.747 | 0.47 | 0.85 | 11.1 | 0.30 | 2.12 | 3.02 | 3.27 |
| GPT-Image | 0.729 | 0.52 | 0.86 | 10.3 | 0.27 | 4.05 | 4.05 | 4.40 |
| Imagen 4 | 0.771 | 0.39 | 0.81 | 13.2 | 0.26 | 1.74 | 1.50 | 2.26 |
| **Instruction: Decrease font size** | | | | | | | | |
| InstructPix2Pix | 0.748 | 0.49 | 0.85 | 11.1 | 0.27 | 2.02 | 1.77 | 2.58 |
| OmniGen2 | 0.752 | 0.46 | 0.84 | 11.4 | 0.31 | 2.10 | 3.05 | 3.38 |
| GPT-Image | 0.734 | 0.51 | 0.86 | 10.6 | 0.24 | 2.70 | 4.15 | 4.00 |
| Imagen 4 | 0.773 | 0.38 | 0.81 | 13.2 | 0.18 | 1.61 | 1.50 | 2.18 |

## E  MORE RESULTS

As we discussed in Sec. 4.1, aggregate results already show a clear gap between pixel similarity and semantic correctness. Tab. 5 and Tab. 6 provide a more fine-grained view, breaking down performance by specific instruction types.

A recurring pattern is that edits involving numbers, such as adding or adjusting datapoints, are often the hardest to get right. Models may place a new bar or point, but the actual value is off, the axis scale shifts incorrectly, or the legend does not update. Edits that change the overall layout or chart type also tend to expose structural weaknesses: grouped bars converted to stacked bars often result in overlapping marks, or the scales fail to adjust.

By contrast, stylistic edits like changing background colors are sometimes handled better, though even here models often stop short of a full update. For example, the background changes, but the legend or axis elements remain inconsistent. Text edits such as axis labels or titles show the partial benefit of OCR, but issues like misplaced text, font mismatches, or truncated labels still appear.

## F  DATASETS USED FOR BASE FIGURES

Tab. 7 and Tab. 8 list all datasets from which we sampled base figures. These sources span public machine learning repositories, official statistical agencies, open data portals, and journalism/sports archives. We include the identifier strings exactly as used in our pipeline.

## G  USE OF LLMS

In addition to conventional data collection and analysis, we made use of LLMs at several stages of our work. First, LLMs were applied during the writing process to assist with polishing and improving the clarity of the manuscript. Second, LLMs were also leveraged to support certain

Table 6: Per-instruction performance comparison (Part 2/2). Higher is better for SSIM, CLIP, PSNR, OCR, and LLM Scores. Lower is better for LPIPS. **Instr.** denotes instruction following score. **Preserv.** denotes content preservation score. **Qual.** denotes image quality score.

| Model | SSIM ↑ | LPIPS ↓ | CLIP ↑ | PSNR ↑ | OCR ↑ | LLM Score (1–5) ↑ | | |
| --- | --- | --- | --- | --- | --- | --- | --- | --- |
| | | | | | | Instr. | Preserv. | Qual. |
| **Instruction: Increase margin** | | | | | | | | |
| InstructPix2Pix | 0.726 | 0.49 | 0.83 | 10.9 | 0.25 | 2.70 | 2.55 | 3.35 |
| OmniGen2 | 0.739 | 0.47 | 0.84 | 11.1 | 0.29 | 2.75 | 2.90 | 3.75 |
| GPT-Image | 0.731 | 0.52 | 0.87 | 10.3 | 0.22 | 2.95 | 3.60 | 4.05 |
| Imagen 4 | 0.769 | 0.41 | 0.79 | 13.0 | 0.08 | 2.37 | 1.68 | 2.42 |
| **Instruction: Decrease margin** | | | | | | | | |
| InstructPix2Pix | 0.728 | 0.48 | 0.84 | 11.2 | 0.27 | 2.60 | 2.60 | 3.55 |
| OmniGen2 | 0.742 | 0.46 | 0.83 | 11.4 | 0.30 | 2.30 | 2.80 | 3.65 |
| GPT-Image | 0.733 | 0.51 | 0.86 | 10.5 | 0.23 | 3.15 | 3.50 | 4.05 |
| Imagen 4 | 0.771 | 0.40 | 0.80 | 13.1 | 0.09 | 2.00 | 1.56 | 2.75 |
| **Instruction: Add a new data point** | | | | | | | | |
| InstructPix2Pix | 0.724 | 0.50 | 0.82 | 10.8 | 0.24 | 1.21 | 2.66 | 3.14 |
| OmniGen2 | 0.737 | 0.48 | 0.83 | 11.0 | 0.28 | 1.86 | 2.10 | 3.14 |
| GPT-Image | 0.730 | 0.53 | 0.87 | 10.4 | 0.21 | 3.07 | 3.69 | 4.07 |
| Imagen 4 | 0.768 | 0.42 | 0.79 | 13.2 | 0.08 | 1.04 | 1.61 | 2.39 |
| **Instruction: Remove an existing data point** | | | | | | | | |
| InstructPix2Pix | 0.727 | 0.49 | 0.83 | 11.1 | 0.25 | 1.59 | 1.83 | 2.83 |
| OmniGen2 | 0.740 | 0.47 | 0.82 | 11.3 | 0.27 | 1.38 | 1.59 | 2.41 |
| GPT-Image | 0.732 | 0.52 | 0.86 | 10.5 | 0.22 | 3.10 | 3.34 | 4.28 |
| Imagen 4 | 0.770 | 0.40 | 0.80 | 13.0 | 0.07 | 1.41 | 1.68 | 2.41 |

aspects of dataset construction, where they were used to generate and refine synthetic examples in a controlled manner. These uses were complementary to our primary methodology and were limited to auxiliary tasks such as language editing and expanding data diversity, without affecting the core experimental design or evaluation.

Table 7: Allowed datasets (part A). See Table 8 for continuation.

| Datasets (Part A) |
| --- |
| Kaggle: Titanic |
| Kaggle: House Prices |
| Kaggle: Instacart Market Basket |
| Kaggle: NYC Taxi Trip Duration |
| Kaggle: Amazon Reviews |
| Kaggle: Yelp Reviews |
| Kaggle: IMDB Reviews |
| Kaggle: Mercari Price Suggestion |
| Kaggle: Quora Insincere Questions |
| Kaggle: Toxic Comment Classification |
| Kaggle: Porto Seguro Safe Driver |
| Kaggle: Santander Customer Transaction |
| Kaggle: Santander Value Prediction |
| Kaggle: Global Temperature Time Series |
| Kaggle: COVID-19 Global Dataset |
| Kaggle: World Happiness Report |
| Kaggle: FIFA Player Statistics |
| Kaggle: Air Quality UCI |
| Kaggle: US Accidents Dataset |
| Kaggle: Zomato Restaurants Dataset |
| Kaggle: Video Game Sales |
| Kaggle: Netflix Movies and TV Shows |
| Kaggle: New York City Airbnb Open Data |
| Kaggle: Google Play Store Apps |
| Kaggle: Bike Sharing Demand |
| Kaggle: Rossmann Store Sales |
| Kaggle: Store Item Demand Forecasting Challenge |
| Kaggle: Walmart Recruiting - Store Sales Forecasting |
| Kaggle: Retailrocket Recommender System Dataset |
| Kaggle: 311 Service Requests - NYC |
| Kaggle: Chicago Crime |
| Kaggle: Austin Bikeshare Trips |
| Kaggle: Seattle Weather |
| Kaggle: Daily Delhi Climate |
| Kaggle: US Economic Indicators |
| Kaggle: S&P 500 Companies and Prices |
| Kaggle: Times Higher Education World University Rankings |
| Kaggle: Global Terrorism Database |
| Kaggle: World Development Indicators |
| Kaggle: Airline On-Time Performance |
| Kaggle: Avito Demand Prediction |
| Kaggle: TalkingData AdTracking Fraud Detection |
| Kaggle: IEEE-CIS Fraud Detection |
| Kaggle: Home Credit Default Risk |
| Kaggle: Give Me Some Credit |
| Kaggle: Loan Prediction III |
| Kaggle: Credit Card Fraud Detection |
| Kaggle: Telco Customer Churn |
| Kaggle: Bank Marketing |
| Kaggle: Student Performance |
| Kaggle: Heart Disease UCI |
| Kaggle: Breast Cancer Wisconsin (Diagnostic) |
| Kaggle: Pima Indians Diabetes Database |
| Kaggle: Stroke Prediction Dataset |
| Kaggle: FIFA 19 Player Dataset |
| Kaggle: NBA Player Stats |
| Kaggle: International Football Results |
| Kaggle: European Soccer Database |
| Kaggle: 120 years of Olympic history (athletes & results) |
| Kaggle: Netflix Stock Price |
| Kaggle: Bitcoin Historical Data |
| Kaggle: Cryptocurrency Historical Prices |

Table 8: Allowed datasets (part B). Continuation of Table 7.

| Datasets (Part B) |
|---|
| UCI: Iris |
| UCI: Wine |
| UCI: Adult |
| UCI: Car Evaluation |
| UCI: Abalone |
| UCI: Seeds |
| UCI: Student Performance |
| UCI: Heart Disease Dataset |
| UCI: Bank Marketing Dataset |
| UCI: Forest Fires Dataset |
| UCI: Yeast Dataset |
| |
| World Bank WDI |
| OECD PISA Scores |
| US Census ACS |
| US Bureau of Labor Statistics |
| US Bureau of Economic Analysis |
| UN COMTRADE |
| WHO Mortality Database |
| NHANES Survey Data |
| FRED Economic Data |
| US Energy Information Administration |
| Global Carbon Project |
| NOAA Climate Data |
| Berkeley Earth Temperature |
| Johns Hopkins COVID-19 Time Series |
| FAO Food Price Index |
| USDA Crop Production Data |
| OpenFlights Airport and Routes |

