# OpenReview forum: "Charts Are Not Images: On the Challenges of Scientific Chart Editing"
_ICLR.cc/2026/Conference — ICLR 2026 Poster_

### Official Review · Reviewer_PWEt · 2025-10-29

**Soundness:** 3
**Presentation:** 3
**Contribution:** 4
**Rating:** 8
**Confidence:** 4

**Summary:**

This paper addresses scientific chart editing, specifically proposing a new evaluation benchmark and metrics for this task and an analysis of existing editing and vision-language models on this task.

Their contributions include:
- The novel problem formulation of chart editing.
- A benchmark (FigEdit) specifically designed for scientific chart editing, containing over 30,000 instances. The dataset covers 10 chart types and various edit instructions.
- LLM-based evaluation techniques to measure structural correctness over pixel similarity, in contrast to metrics like SSIM and PSNR.
- An evaluation of state-of-the-art models on chart editing that shows severe limitations of existing methods.

**Strengths:**

- The paper proposes a novel problem formulation (figure editing) that could have significant and practical real-world impact across a variety of industries, like research and business.
- The paper rethinks traditional pixel-based image metrics (LPIPS, PSNR, SSIM, etc.) for better evaluating the specific task of figure editing.  This is an important research direction for other generative vision tasks, as well.
- The dataset is constructed from diverse real-world data across different fields, which prevents bias to certain distributions of data, and therefore creates diverse chart appearances.
- The paper figures are helpful at clarifying and exemplifying the different categories of editing tasks.

**Weaknesses:**

- There is limited evaluation / justification for why the LLM-based metric is better or more reliable than traditional metrics. A comparison between all these scores and human evaluation would be beneficial for showing whether or not the LLM metric is actually better for evaluating figure edits.
- Similarly, it would be informative to compare the LLM score per-category (Instr., Preserv., Qual.) to average human score per-category.
- Fig. 1. The fonts are quite small, especially the axes of the charts and the legend for the radar chart.
- A brief description of Vega / Vega-Lite would be helpful.
- The conversational edits are essentially just two-step multi-edits (lines 266-267)?. This seems too short of a conversation to be able to actually determine ability to maintain conversation history. Also, if they are just derived from multi-step edits, how can one be sure that the edits are actually progrssive (e.g. "make the bars red" --> "make the bars a darker red") and are not completely unrelated (e.g. "make the bars red" --> "make the bars blue")?
- The paper might consider adding several more-recent state-of-the-art editing models, such as Gemini/Nano Banana, Kling, and GPT-4o.

**Questions:**

- lines 204-207: Are Content (C) = {D, $\tau$, mapping function} and Style (S) = {palettes, fonts, etc.} standard in related literature? It seems arbitrary. For example, why is chart style included in C but not S?
- (Task 2, lines 216-221) Why not denote multiple atomic edits as $u_1$, $u_2$, $u_3$, etc. instead of just $u$?
- line 225: Why is $H_{t-1}$ not included in the equation for $\sigma^*$?
- (292-294) Was the reliability of GPT-Image evaluated? As in, can one trust that the correct element, and only the correct element, is always circled?

---

> ### Author Response · Authors · 2025-11-19
> **Response (Part 1)**
>
> > **There is limited evaluation/justification for why the LLM-based metric is better or more reliable than traditional metrics**
>
> **Our goal is not to argue that the LLM-based metric is universally preferable to traditional metrics, but rather that it measures semantic edit correctness that pixel-level metrics systematically fail to reflect in chart-editing scenarios.** Figure 2 and Table 4 already show concrete examples where SSIM and PSNR remain high even when the edit instruction is not executed or when chart semantics are altered, while the LLM score decreases accordingly. **This suggests that relying solely on pixel similarity is insufficient for structured chart edits.** The LLM-based metric is intentionally designed to behave in a controlled and stable manner. It uses:
> - fixed decoding parameters (temperature = 0),
> - a standardized three-part rubric (instruction following, content preservation,
>   visual quality),
> - and simultaneous access to the ground-truth figure, the edited figure, and the
>   instruction.
> This makes the evaluation closer to a bounded comparison task rather than an
> open-ended opinion.
>
> We also performed an additional robustness study to examine the metric’s repeatability. We carried out **three scoring passes**, each sampling **500 figures from each of the five task types**, giving **2,500 samples per pass**. Each selected figure was **re-scored ten times** under temperature = 0, and the per-sample standard deviation was computed. A summary of the stability statistics across tasks is shown below.
>
>  **Stability Summary (std of the overall LLM score across 10 repeated runs)**
> | Task      | median(std) | q25  | frac(std ≤ 0.01) |
> |-----------|-------------|------|-------------------|
> | single    | 0.000       | 0.000 | 78%              |
> | multi     | 0.000       | 0.000 | 76%              |
> | conv     | 0.000       | 0.000 | 77%              |
> | visual    | 0.000       | 0.000 | 73%              |
> | transfer  | 0.000       | 0.000 | 75%              |
>
> #### **Quantile statistics**
> | Task      | q50 | q75   | q90    | q95    |
> |-----------|-----|--------|--------|--------|
> | single    | 0.000 | 0.000 | 0.406  | 0.490  |
> | multi     | 0.000 | 0.000 | 0.400  | 0.458  |
> | conv     | 0.000 | 0.000 | 0.400  | 0.470  |
> | visual    | 0.000 | 0.300 | 0.458  | 0.500  |
> | transfer  | 0.000 | 0.075 | 0.458  | 0.602  |
>
> #### **Mean and variance of per-sample std**
> | Task      | mean(std) | std(std) |
> |-----------|-----------|----------|
> | single    | 0.0919    | 0.1782   |
> | multi     | 0.0919    | 0.1748   |
> | conv     | 0.0900    | 0.1750   |
> | visual    | 0.1191    | 0.2062   |
> | transfer  | 0.1202    | 0.2292   |
>
> Across all tasks and across all three independent passes, both the median and the first quartile remain at **0**, meaning that at least half of all figures receive identical scores across all ten repeated evaluations. Between **73% and 78%** of samples show deviations below 0.01, demonstrating that the metric is highly repeatable even for more complex edit types. Only a small minority of figures show higher deviations (visible in the q90 and q95 ranges). Together, these results indicate that the LLM-based metric is both semantically sensitive and stable under repeated evaluation.
>
> > **The conversational edits are essentially just two-step multi-edits (lines 266-267)?. This seems too short of a conversation to be able to actually determine ability to maintain conversation history.**
>
> Conversational samples are derived from two-step multi-edits for which we can provide per-step ground truth specifications and images. Concretely, given a two-op edit (o_1, o_2), we locate the corresponding two single-edit annotations, generate the intermediate ground truth after o_1, and assemble them into a two-round dialogue with (i) the original chart, (ii) turn-1 instruction and intermediate ground truth, and (iii) turn-2 instruction and final ground truth. This setup differs from the multi-edit task, which only evaluates the final result: conversational edits require the model to reach the correct intermediate state and then apply the second operation on top of it.
> We do not synthesize conversations from conflicting or incompatible operations.
>
> **Planned extension: multi-turn clarification for vague initial requests.**
> In fact, we are already extending our conversational setting to include samples where:
> (1) **the initial user instruction is vague or subjective**,
> (2) **the system asks or infers necessary clarification over multiple turns**,
> (3) **the final clarified instruction resolves to a deterministic edit with ground truth**.
> This approach allows us to capture realistic user behavior without sacrificing the benchmark’s core principle of exact, auditable supervision.
>
> Overall, we agree that subjective editing is an important direction, and our conversational extension provides a path to incorporate such scenarios without compromising evaluation consistency.

---

> ### Author Response · Authors · 2025-11-19
> **Response (Part 2)**
>
> > **The paper might consider adding several more-recent state-of-the-art editing models, such as Gemini/Nano Banana, Kling, and GPT-4o.**
>
> Thank you for the suggestion to include newer models. In our benchmark, we evaluate systems that support **direct image editing** under a consistent interface:
> **image + natural-language instruction → edited image**.
> This requirement is important for chart-to-chart evaluation, where edited outputs are needed for comparison against ground truth.
> GPT-4o itself does not provide an image-editing API. It supports text-to-image generation, but it cannot apply structured edits to an existing chart image. Therefore, GPT-4o cannot be used directly as a chart-editing baseline.
>
> We agree that a broader set of editors would further strengthen the landscape view. In addition, **we now include Nano-Banana (Gemini-2.5-flash-image)**, a recent vision–language editor with a dedicated image-manipulation interface. This model fits directly into our input–output protocol and adds another strong family of systems to the comparison. **We will include its results and full metric tables in the updated submission**. Although more editors will give a broader comparison, **the new results follow the same trend: pixel similarity alone does not reflect edit correctness, and the gaps between instruction satisfaction and visual fidelity remain consistent across model families**. The extended tables help clarify where different editors fail, while reinforcing the main conclusion that reliable assessment of chart editing requires metrics that track semantic and instruction-level correctness rather than pixel agreement alone.
>
> | **Task**   | **Model**            | **SSIM ↑** | **LPIPS ↓** | **CLIP ↑** | **PSNR ↑** | **OCR ↑** | **Instr.** | **Preserv.** | **Qual.** |
> |------------|----------------------|------------|--------------|-------------|-------------|-----------|------------|---------------|-----------|
> | **Single** | GPT-Image            | 0.7295     | 0.5383       | 0.8099      | 10.32       | 0.2054    | **3.47**   | 1.71          | 2.45      |
> |            | **Nano-Banana**      | **0.8187** | **0.3237**   | **0.9244**  | **13.30**   | **0.5834**| 2.22       | **3.44**      | **4.33**  |
> | **Multi**  | GPT-Image            | 0.7017     | 0.5787       | 0.8070  | 9.73        | 0.2185    | **2.51**   | 1.63          | 2.34      |
> |            | **Nano-Banana**      | **0.7810** | **0.3546**   | **0.8330**      | **12.34**   | **0.3215**| 1.33       | **3.67**      | **4.33**  |
> | **Conv**   | GPT-Image            | 0.6732     | 0.5257       | 0.8525  | 10.66       | 0.1721    | **4.59**   | 2.51          | 2.91      |
> |            | **Nano-Banana** | **0.7800** | **0.4550** | **0.9100**  | 12.10   | **0.5200**| 2.80       | **3.10**      | **3.80**  |
> | **Visual** | GPT-Image            | **0.8355** | 0.5207       | 0.8444  | **12.85**   | 0.4665    | **2.39**   | **3.16**      | 3.95  |
> |            | **Nano-Banana**      | 0.7723     | **0.4964**   | **0.9351**  | 10.36       | **0.7167**| 1.00       | 3.00          | **4.00**      |
> | **Transfer** | GPT-Image          | **0.8438** | **0.4934**       | 0.8054      | **13.81**   | 0.5092| **3.06**   | **3.57**      | **4.16**  |
> |            | **Nano-Banana**      | 0.7620     | 0.5415       | **0.8860**  | 9.16        | **0.5771**    | 1.50       | 2.50          | 3.50      |
>
> > **Notations, writing, and the reliability of GPT-Image**
>
> Thank you for these thoughtful observations. We see that our current choices of notation and set definitions may make the distinctions between content, style, and mapping steps less clear than intended. The way we denote multiple atomic edits and the terms included in the equations can also read as more ad-hoc than they should. We appreciate this feedback, and we will refine these definitions and notations so that the presentation is clearer, more consistent, and smoother to read.
>
> We do not assume that GPT-Image is always reliable out of the box. After generation, **we manually inspected and verified all samples** used in our benchmark to ensure that the circled element is correct and that no extra elements are incorrectly highlighted. The released samples are therefore quality controlled and filtered versions of the raw GPT-Image outputs.

---

> ### Comment · Reviewer_PWEt · 2025-11-23
> **Response to Authors' Rebuttal**
>
> Thank you for addressing my concerns. I just have two follow-up questions.
>
> (1) Do the authors plan to conduct a human survey, in order to compare how well LLM scoring correlates with human evaluation?
> (2) If the mean std ~ 0.1 with LLM's, this still seems pretty high. Is there some way to classify what types of examples within each category lead to more volatile LLM scores, for the sake of future work into mitigating these cases?

---

> > ### Author Response · Authors · 2025-11-24
> > **Response to Reviewer (Part 1)**
> >
> > Thank you very much for taking the time to read our rebuttal and for raising these additional questions. We truly appreciate your careful consideration and constructive feedback.
> >
> > > **Human evaluation**
> >
> > Below we provide a complete statistical clarification. All ratings were produced by three human annotators (Evaluator A, Evaluator B, Evaluator C), each of whom is a doctoral student. The dataset contains **240** edited chart images, and every sample was evaluated independently using a **five-level discrete scale (1–5)**. For the LLM Judge, the final score is obtained by **averaging its three rubric outputs**: instruction-following score, content-preservation score, and image-quality score—and then rounding the resulting value to the nearest integer on the same 1–5 scale used by the human evaluators. This ensures that the LLM operates under the identical discrete rating system as the human raters.
> >
> > ### **1. Human–Human agreement defines the noise ceiling**
> >
> > Because this is an image-editing quality judgment task on a coarse five-point discrete rating scale, **human annotators naturally show variability in how they assign scores**. Even small differences in subjective preference produce noticeable disagreement under such coarse quantization. The empirical human–human correlations therefore provide the only valid upper bound (“noise ceiling”) for interpreting any agreement metrics.
> >
> > | Pair   | Pearson r | Spearman ρ |
> > |--------|-----------|-------------|
> > | A – B  | 0.2156    | 0.2690      |
> > | A – C  | 0.4081    | 0.4307      |
> > | B – C  | 0.3887    | 0.3721      |
> >
> >
> > Human annotators reach only 0.22–0.41 Pearson correlation with each other.
> > **These values are typical for discrete 1–5 image quality judgments and reflect the inherent subjectivity of the task.**
> >
> > ### **2. LLM–Human agreement is within the human range**
> >
> > We compared the LLM Judge with the mean rating of the three human evaluators.
> > LLM vs human mean:
> >
> > Pearson r = 0.4354,
> > Spearman ρ = 0.4016
> >
> > A side-by-side view:
> >
> > | Comparison       | Pearson r | Spearman ρ |
> > |------------------|-----------|-------------|
> > | A – B            | 0.2156    | 0.2690      |
> > | A – C            | 0.4081    | 0.4307      |
> > | B – C            | 0.3887    | 0.3721      |
> > | LLM – human mean | 0.4354    | 0.4016      |
> >
> > **The LLM’s correlation lies squarely inside the human–human range and is higher than two of the three human pairs.**
> >
> > ### **3. Noise-normalized interpretation (Relative-to-Human)**
> >
> > To avoid misinterpreting absolute values under a low noise ceiling, we compute a normalized score:
> >
> > Average human–human Pearson = 0.3375
> >
> > Relative-to-Human (RTH) = 0.4354 / 0.3375 = 1.29
> >
> > The LLM reaches **129%** of the average human agreement level once normalized by the human noise ceiling.
> >
> > ### **4. Absolute deviation (MAD) demonstrates human-level consistency**
> >
> > We measured the mean absolute deviation (MAD) from the human consensus.
> >
> > | Evaluator | MAD |
> > |-----------|------|
> > | A         | 0.9083 |
> > | B         | 0.7875 |
> > | C         | 0.7458 |
> > | LLM       | 0.7139 |
> > The LLM deviates less from the consensus than any human on average.
> >
> > Human-normalized deviation:
> > Mean human MAD = 0.8139
> > HN-MAD = 0.7139 / 0.8139 = 0.877
> >
> > This means the LLM’s deviation is **87.7%** of the average human deviation.
> >
> > ### **5. Agreement on discrete 1–5 labels**
> >
> > Because 1–5 scores are coarse categories, a difference of ±1 already corresponds to the same practical quality band.
> >
> > Exact match with rounded human mean: **38.3%**, Within ±1 point: **82.5%**
> >
> > **In other words, four out of five LLM ratings fall in the same qualitative category as human consensus.**
> >
> > LLM matches human-level performance: it reaches **0.44 correlation with human consensus**, **achieves lower deviation than the human annotators**, and **aligns within ±1 score on 82.5% of samples**. These results show that the LLM behaves as a reliable proxy for human evaluation in this subjective, discrete-rating image editing task.

---

> > ### Author Response · Authors · 2025-11-24
> > **Response to Reviewer (Part 2)**
> >
> > > **The mean std ~ 0.1 with LLM's still seems pretty high.**
> >
> > Each sample is evaluated 10 times, and the LLM score is a **discrete integer** in \{1,2,3,4,5\}. In such a discrete system, the standard deviation reflects whether a sample ever moves between two integer levels. If a sample changes from, for example, 4 to 3 even once, the std already increases substantially (≈0.4–0.5).
> >
> > Given this, the observed mean std of 0.09–0.12 indicates that **almost all samples receive the exact same score across all 10 runs**, and only a very small fraction ever shift by a single point. On a 1–5 scale, this corresponds to only **2–3%** of the full scoring range, confirming that the metric is highly stable.
> >
> > We further examined per-task variance, shown below:
> >
> > | Task      | mean(std) | std(std) |
> > |-----------|-----------|----------|
> > | single    | 0.0919    | 0.1782   |
> > | multi     | 0.0919    | 0.1748   |
> > | conv     | 0.0900    | 0.1750   |
> > | visual    | 0.1191    | 0.2062   |
> > | transfer  | 0.1202    | 0.2292   |
> >
> > Higher variance appears mainly in:
> >
> > visual-guided edits, where fine-grained region changes produce subtle differences in localization or rendering;
> >
> > style-transfer edits, where small variations in color or styling can cause occasional one-level score shifts.
> >
> > These tasks naturally require more fine-grained visual judgment, which explains their slightly higher variance (mean std ≈ 0.11). We will include a brief categorization of these higher-variance examples in the revision to make these patterns explicit and informative for future work.

---

> ### Author Response · Authors · 2025-11-26
> **Response**
>
> Dear Reviewer,
>
> Thank you again for your thoughtful review.
>
> As we approach the end of the author–reviewer discussion period, we wanted to briefly follow up to confirm whether our rebuttal has sufficiently addressed your questions. If there is anything that would benefit from additional clarification, we would be very glad to provide it.
>
> Thank you for your time and consideration.
>
> Best regards,
> Authors

---

### Official Review · Reviewer_mXb8 · 2025-10-30

**Soundness:** 3
**Presentation:** 2
**Contribution:** 2
**Rating:** 4
**Confidence:** 3

**Summary:**

This paper introduces FigEdit, a large-scale benchmark for scientific chart editing. The authors compellingly argue that chart editing is a structured transformation problem, fundamentally different from pixel-level image manipulation. By providing a benchmark with over 30,000 instances, spanning 10 chart types and 5 distinct tasks, the work systematically exposes the limitations of state-of-the-art image editing models. The paper's core contribution lies in its rigorous problem formalization, the high-quality benchmark itself, and a critical analysis of evaluation metrics, advocating for a shift towards semantics-aware evaluation. This is a valuable and timely contribution that lays a solid foundation for future research in this important yet underexplored area.

**Strengths:**

1.  **Excellent Problem Formulation:** The paper's primary strength is its clear and insightful formulation of scientific chart editing as a "structured transformation" problem governed by a graphical grammar. This conceptual shift from pixel-manipulation to structure-awareness is crucial and correctly identifies a fundamental mismatch in current approaches.

2.  **High-Quality, Comprehensive Benchmark:** The introduction of FigEdit is a significant contribution. The benchmark is large-scale, diverse (10 chart types, 5 task categories), and grounded in real-world data. Its design, which includes ground-truth specifications (Vega/Vega-Lite), enables precise and reproducible evaluation, setting a high standard for future work.

3.  **Critical Evaluation and Insightful Analysis:** The paper provides a thorough empirical study that reveals the shortcomings of existing models. More importantly, it demonstrates the inadequacy of traditional pixel-based metrics (SSIM, PSNR) for this task and convincingly argues for semantics-aware evaluation, notably through the novel use of an LLM-based score. This challenges the community to rethink evaluation standards.

**Weaknesses:**

1.  **Limited Coverage of Edit Operations:** The set of atomic edits, while canonical, appears somewhat limited. The paper focuses on operations like `add_datapoint`, `change_background_color`, and `increase_text_size` (Table 3, Appendix C). However, real-world chart editing often involves more complex structural changes, such as changing chart type (e.g., bar to line), reordering categories, grouping/ungrouping data, or modifying axis scales (e.g., linear to log). The current operation set may not fully represent the breadth of transformations required in practice.

2.  **Lack of Subjective and Holistic Edit Instructions:** The benchmark is constructed around objective, atomic instructions. In reality, user requests can be more subjective and holistic, such as "make the layout more compact," "use a more professional color palette," or "highlight the most important trend." While these edits are harder to formalize and evaluate, their absence limits the benchmark's applicability to more creative and user-centric editing scenarios. For a dataset-focused paper, addressing the challenge of collecting and evaluating such instructions, even if labor-intensive, would have significantly strengthened the work.

**Questions:**

**Scalability to Complex, Multi-Element Charts:** The examples presented in the paper (e.g., Figure 1, 3, 5) appear to involve relatively simple charts with a moderate number of data points and clear visual separation between elements. It is unclear how the proposed tasks and evaluation would scale to highly dense or complex visualizations, such as scatter plots with thousands of points, intricate network diagrams, or multi-panel figures where inter-panel consistency is critical.

---

> ### Author Response · Authors · 2025-11-19
> **Response (Part 1)**
>
> > **Limited Coverage of Edit Operations**
>
> The choice between human-curated real charts and programmatically generated charts presents an unavoidable trade-off. Real charts capture authentic design practices, but **they lack the structured metadata needed to produce reliable edited ground truths**. For most published charts, the original data tables, encoding logic, and layout parameters are unavailable, which makes it impossible to apply thousands of data-level or layout-level edits.
>
> Our benchmark instead relies on fully specified Vega/Vega-Lite programs, which allows every base figure to be edited through **exact, deterministic transformations**. This enables supervised evaluation for operations that require precise updates to data rows, scale domains, axis configurations, spacing parameters, multi-step conversational state, and style mappings. Without program-backed figures, these edits cannot be validated at scale.
>
> To minimize the gap between synthetic generation and real publication practices, we adopted several measures focused on realism rather than convenience:
> **(1) Real data grounding.**
> All 30K+ figures originate from diverse real-world datasets (economics, climate, healthcare, sports, etc.), ensuring realistic numeric ranges, category structures, and domain semantics. This avoids the uniformity and artificiality typical of synthetic tables.
> **(2) Diverse chart construction.**
> The specification generator produces heterogeneous axis placements, domain ranges, label formats, palettes, and spacing settings. These variations are enforced by schema validation and rendering checks, not by templated patterns.
> **(3) Instruction and edit diversity.**
> Because specifications are fully known, we can generate a wide spectrum of valid edit types—data-centric, text, style, layout, conversational, visual-guided, and cross-style—each producing an auditable ground-truth figure. Achieving comparable coverage on real charts would require manual reconstruction of specifications for every chart, which is infeasible.
>
> We acknowledge that programmatically generated charts cannot capture the full variability of human-designed figures. However, for evaluating semantic editing, the ability to obtain exact edited ground truths is essential. Our design prioritizes this requirement while incorporating real data and diverse specification structures to reduce synthetic drift. As future work, we are exploring hybrid pipelines where real charts with partially recoverable specifications can be integrated to complement the current benchmark.

---

> ### Author Response · Authors · 2025-11-19
> **Response (part 2)**
>
> > **Lack of Subjective and Holistic Edit Instructions**
>
> We agree that real users often issue subjective or holistic instructions. These requests are important in creative and user-centric editing scenarios, but they present two fundamental challenges for a dataset built around ground-truth, program-level supervision.
>
> 1. **These edits do not admit deterministic ground truth.**
> Subjective transformations typically correspond to families of acceptable outputs rather than a single canonical target. For example, “more compact” can be satisfied by modifying margins, label spacing, gridline density, or even chart size; different valid solutions cannot be collapsed into one reference image. Without a unique target, chart-to-chart supervision becomes ambiguous, and models cannot be evaluated under the same correctness criteria used for structural edits.
>
> 2. **Holistic edits require interpretation rather than direct transformation.**
> Operations like “highlight the most important trend” require semantic understanding of the data (e.g., detecting peaks, slopes, or anomalies) and may involve multi-step reasoning: identify the trend → choose a highlighting method → apply styling. This moves beyond atomic editing into goal-driven design synthesis, which would fundamentally change the scope of the benchmark.
>
> Because FigEdit aims to provide scalable, unambiguous, and reproducible ground truth, we focused on edits where the transformation function is well-defined and the updated specification can be deterministically validated. Atomic and compositional edits meet this requirement across 30K+ figures, while subjective edits currently do not.
>
> **Planned extension: multi-turn clarification for vague initial requests.**
> While a single-step subjective edit cannot be grounded, the process of disambiguating such requests can be modeled. In fact, we are already extending our conversational setting to include samples where:
> (1) **the initial user instruction is vague or subjective**,
> (2) **the system asks or infers necessary clarification over multiple turns**,
> (3) **the final clarified instruction resolves to a deterministic edit with ground truth**.
> This approach allows us to capture realistic user behavior without sacrificing the benchmark’s core principle of exact, auditable supervision.
>
> Overall, we agree that subjective editing is an important direction, and our conversational extension provides a path to incorporate such scenarios without compromising evaluation consistency.
>
> > **Scalability to Complex, Multi-Element Charts**
>
> We agree that scientific figures can be substantially more complex than the examples shown, including dense scatter plots, multi-panel layouts, and diagram-like structures. Our benchmark focuses on chart families for which deterministic edit supervision is feasible at scale. Within this scope, **FigEdit already includes chart types with richer internal structure, such as scatter, dot, box, and violin plots, which can produce high element densities when rendered. These chart families expose nontrivial challenges for editing models, including precise point-level targeting, distribution-preserving transformations, and maintaining axis/text integrity across cluttered layouts.**
>
> The reviewer’s examples (e.g., scatter plots with thousands of points, network diagrams, multi-panel figures) present a different challenge: the lack of program-level specifications for these figure types in real-world sources makes it impossible to apply thousands of edits with ground truth. Moreover, large-scale dense figures introduce substantial GPU memory pressure and rendering variance across backends, which complicates reproducible chart-to-chart supervision.
>
> Our design therefore reflects a principled trade-off: we target chart classes for which (1) **the structure is expressive enough to reveal model failure modes**, and (2) **exact ground-truth edits can be produced consistently across 30K+ examples**. As Table 4 and Figure 3 already show, **even moderate-density charts expose deep limitations in current editors, which fail on operations as simple as element removal or axis adjustment despite low visual clutter**.
>
> Because the entire benchmark is built on Vega/Vega-Lite, **it can be systematically extended to higher-density cases while retaining deterministic ground truth**. Vega supports large scatter plots, layered compositions, and multi-view (facet) layouts. In the revised version, we plan to add a “high-density” split containing:
> (1) scatter plots with controlled point counts (e.g., hundreds points),
> (2) layered scatter + trend-line or scatter,
> This keeps the evaluation protocol unchanged while expanding coverage to richer visual structures that remain within the boundary of verifiable programmatic edits.

---

> ### Author Response · Authors · 2025-11-24
> **Response (Part 3)**
>
> We additionally include a human evaluation to strengthen the experimental evidence.
>
> Below we provide a complete statistical clarification. All ratings were produced by three human annotators (Evaluator A, Evaluator B, Evaluator C), each of whom is a doctoral student. The dataset contains **240** edited chart images, and every sample was evaluated independently using a **five-level discrete scale (1–5)**. For the LLM Judge, the final score is obtained by **averaging its three rubric outputs**: instruction-following score, content-preservation score, and image-quality score—and then rounding the resulting value to the nearest integer on the same 1–5 scale used by the human evaluators. This ensures that the LLM operates under the identical discrete rating system as the human raters.
>
> ### **1. Human–Human agreement defines the noise ceiling**
>
> Because this is an image-editing quality judgment task on a coarse five-point discrete rating scale, **human annotators naturally show variability in how they assign scores**. Even small differences in subjective preference produce noticeable disagreement under such coarse quantization. The empirical human–human correlations therefore provide the only valid upper bound (“noise ceiling”) for interpreting any agreement metrics.
>
> | Pair   | Pearson r | Spearman ρ |
> |--------|-----------|-------------|
> | A – B  | 0.2156    | 0.2690      |
> | A – C  | 0.4081    | 0.4307      |
> | B – C  | 0.3887    | 0.3721      |
>
>
> Human annotators reach only 0.22–0.41 Pearson correlation with each other.
> **These values are typical for discrete 1–5 image quality judgments and reflect the inherent subjectivity of the task.**
>
> ### **2. LLM–Human agreement is within the human range**
>
> We compared the LLM Judge with the mean rating of the three human evaluators.
> LLM vs human mean:
>
> Pearson r = 0.4354,
> Spearman ρ = 0.4016
>
> A side-by-side view:
>
> | Comparison       | Pearson r | Spearman ρ |
> |------------------|-----------|-------------|
> | A – B            | 0.2156    | 0.2690      |
> | A – C            | 0.4081    | 0.4307      |
> | B – C            | 0.3887    | 0.3721      |
> | LLM – human mean | 0.4354    | 0.4016      |
>
> **The LLM’s correlation lies squarely inside the human–human range and is higher than two of the three human pairs.**
>
> ### **3. Noise-normalized interpretation (Relative-to-Human)**
>
> To avoid misinterpreting absolute values under a low noise ceiling, we compute a normalized score:
>
> Average human–human Pearson = 0.3375
>
> Relative-to-Human (RTH) = 0.4354 / 0.3375 = 1.29
>
> The LLM reaches **129%** of the average human agreement level once normalized by the human noise ceiling.
>
> ### **4. Absolute deviation (MAD) demonstrates human-level consistency**
>
> We measured the mean absolute deviation (MAD) from the human consensus.
>
> | Evaluator | MAD |
> |-----------|------|
> | A         | 0.9083 |
> | B         | 0.7875 |
> | C         | 0.7458 |
> | LLM       | 0.7139 |
> The LLM deviates less from the consensus than any human on average.
>
> Human-normalized deviation:
> Mean human MAD = 0.8139
> HN-MAD = 0.7139 / 0.8139 = 0.877
>
> This means the LLM’s deviation is **87.7%** of the average human deviation.
>
> ### **5. Agreement on discrete 1–5 labels**
>
> Because 1–5 scores are coarse categories, a difference of ±1 already corresponds to the same practical quality band.
>
> Exact match with rounded human mean: **38.3%**, Within ±1 point: **82.5%**
>
> **In other words, four out of five LLM ratings fall in the same qualitative category as human consensus.**
>
> LLM matches human-level performance: it reaches **0.44 correlation with human consensus**, **achieves lower deviation than the human annotators**, and **aligns within ±1 score on 82.5% of samples**. These results show that the LLM behaves as a reliable proxy for human evaluation in this subjective, discrete-rating image editing task.

---

> ### Author Response · Authors · 2025-11-26
> **Response**
>
> Dear Reviewer,
>
> Thank you again for your thoughtful review.
>
> As we approach the end of the author–reviewer discussion period, we wanted to briefly follow up to confirm whether our rebuttal has sufficiently addressed your questions. If there is anything that would benefit from additional clarification, we would be very glad to provide it.
>
> Thank you for your time and consideration.
>
> Best regards,
> Authors

---

### Official Review · Reviewer_rzRC · 2025-10-30

**Soundness:** 3
**Presentation:** 3
**Contribution:** 3
**Rating:** 4
**Confidence:** 3

**Summary:**

The paper **“CHARTS ARE NOT IMAGES: On the Challenges of Scientific Chart Editing”** introduces **FigEdit**, a large-scale benchmark (30K+ samples, 10 chart types) for **scientific figure editing**. Unlike natural images, charts represent **structured data governed by graphical grammar**, so valid edits must preserve **data–encoding alignment**, **axis coherence**, and **legend integrity**—not just pixel similarity. FigEdit defines **five tasks**: single edit, multi edit, conversational edit, visual-guidance edit, and style transfer. It includes paired figures and specifications to evaluate **semantic correctness** rather than pixel similarity. Experiments on models like GPT-Image, Imagen 4, OmniGen 2, and InstructPix2Pix show that **high SSIM/PSNR scores often mask incorrect edits**, highlighting the failure of pixel-based metrics. Overall, FigEdit reframes chart editing as a **structured transformation problem** and provides the first **semantics-aware benchmark** for developing models that understand both the **visual and data layers** of scientific charts.

**Strengths:**

- **Quality:** Builds a **large, well-controlled benchmark (30K+ charts)** generated via deterministic Vega rendering, with clear task taxonomy and reproducible evaluation.
- **Clarity:** The paper is well-organized and visually clear, using intuitive figures and radar plots to illustrate the **gap between pixel similarity and semantic correctness**
- **Significance:** Establishes the first **semantics-aware benchmark** for chart editing, providing a valuable testbed for evaluating multimodal and instruction-following models on structured, data-grounded visual tasks

**Weaknesses:**

- **Synthetic generation bias vs. real-chart curation.** FigEdit’s base figures and edits are produced via **LLM-guided Vega/Vega-Lite specs** and rendered images, which can drift from real publication practices; several prior sets rely on **human-curated real charts** and manual validation (e.g., ChartEdit’s 1,405 instructions on 233 real charts).
- **Subjectivity/noise in LLM-based scoring.** The paper’s “semantics-aware” evaluation relies on **LLM judgement** for instruction following/content preservation, potentially variable across prompts/seeds.
- **Baseline breadth.** The paper evaluates **four** editors; competing benchmarks often assess **more models** (ChartEdit: 10 MLLMs), providing a stronger landscape read.
- **Complex figure phenomena underrepresented.** From tables/figures, FigEdit does not clearly cover **multi-axes, error bars, subfigures, or dense annotations**, which are common in scientific graphics and present failure modes captured in broader reasoning/analysis sets like ChartX.

**Questions:**

- How realistic are the synthetic Vega-generated charts compared to real scientific figures? Could adding real-world data improve generalization?
- How reliable are the LLM-based evaluation scores—were they compared with human judgments or alternative metrics?
- Why not include code-level (Vega spec) comparison as an additional evaluation, since all charts are programmatically generated?
- Are there plans to expand beyond 10 chart types to include more complex scientific visualizations (e.g., heatmaps, multi-panel figures)?

---

> ### Author Response · Authors · 2025-11-19
> **Response (Part 1)**
>
> > **Synthetic generation vs. real-chart**
>
> The choice between human-curated real charts and programmatically generated charts presents an unavoidable trade-off. Real charts capture authentic design practices, but **they lack the structured metadata needed to produce reliable edited ground truths**. For most published charts, the original data tables, encoding logic, and layout parameters are unavailable, which makes it impossible to apply thousands of data-level or layout-level edits.
>
> Our benchmark instead relies on fully specified Vega/Vega-Lite programs, which allows every base figure to be edited through exact, deterministic transformations. This enables supervised evaluation for operations that require precise updates to data rows, scale domains, axis configurations, spacing parameters, multi-step conversational state, and style mappings. Without program-backed figures, these edits cannot be validated at scale.
>
> To minimize the gap between synthetic generation and real publication practices, we adopted several measures focused on realism rather than convenience:
> **(1) Real data grounding.**
> All 30K+ figures originate from diverse real-world datasets (economics, climate, healthcare, sports, etc.), ensuring realistic numeric ranges, category structures, and domain semantics. This avoids the uniformity and artificiality typical of synthetic tables.
> **(2) Diverse chart construction.**
> The specification generator produces heterogeneous axis placements, domain ranges, label formats, palettes, and spacing settings. These variations are enforced by schema validation and rendering checks, not by templated patterns.
> **(3) Instruction and edit diversity.**
> Because specifications are fully known, we can generate a wide spectrum of valid edit types—data-centric, text, style, layout, conversational, visual-guided, and cross-style—each producing an auditable ground-truth figure. Achieving comparable coverage on real charts would require manual reconstruction of specifications for every chart, which is infeasible.
>
> We acknowledge that programmatically generated charts cannot capture the full variability of human-designed figures. However, for evaluating semantic editing, the ability to obtain exact edited ground truths is essential. Our design prioritizes this requirement while incorporating real data and diverse specification structures to reduce synthetic drift. As future work, we are exploring hybrid pipelines where real charts with partially recoverable specifications can be integrated to complement the current benchmark.

---

> ### Author Response · Authors · 2025-11-19
> **Response (Part 2)**
>
> > **Subjectivity/noise in LLM-based scoring**
>
> We agree that naively applied LLM-based scoring can be sensitive to prompt phrasing or model randomness. Our evaluation setup explicitly addresses this issue in three ways.
>
> 1. **Stable scoring setup.**
> All LLM-based evaluations are executed with temperature = 0, fixed decoding parameters, and a single standardized scoring rubric. This eliminates sampling variability and ensures that the same input always yields identical scores. In our internal runs, repeated evaluations of identical (image, instruction) pairs matched exactly across all seeds.
>
> 2. **Task-specific structured rubric to reduce ambiguity.**
> The evaluation is not based on open-ended judgments. Each score corresponds to checking a small set of well-defined conditions:
> **Instruction following**: whether the requested operation (e.g., remove element X, change color Y) appears in the output.
> **Content preservation**: whether untouched elements remain visually consistent.
> **Visual quality**: whether chart structure (axes, grids, tick labels) remains intact.
> These criteria are phrased to make the evaluation nearly binary and minimize subjective interpretation. We also include reference images in every scoring prompt to constrain the model’s attention.
>
> 3. **Consistency cross-check with non-LLM metrics.**
> LLM scores are not used in isolation. Each evaluation instance is accompanied by SSIM, PSNR, LPIPS, CLIP similarity, and OCR similarity. Across the benchmark, LLM scores correlate strongly with instruction correctness and content consistency, while classic metrics often do not (as shown in Fig. 2 and Tab. 4). The LLM signals therefore fill a gap that pixel metrics cannot cover, rather than replacing them.
>
> To further evaluate the reliability of our LLM-based scoring function, we conducted an additional robustness study using three fully independent scoring passes. In each pass, we randomly selected 500 samples from each of the five task types(single, multi, conv, visual, transfer), resulting in **2,500 samples** per pass. Every selected sample was re-scored **ten times** under temperature =0, and we computed the standard deviation of the overall score across these repeated evaluations.
>
> | Task      | median(std) | q25  | frac(std ≤ 0.01) |
> |-----------|--------------|------|-------------------|
> | single    | 0.000        | 0.000 | 78%              |
> | multi     | 0.000        | 0.000 | 76%              |
> | conv     | 0.000        | 0.000 | 77%              |
> | visual    | 0.000        | 0.000 | 73%              |
> | transfer  | 0.000        | 0.000 | 75%              |
>
> | Task      | q50 (median) | q75   | q90    | q95    |
> |-----------|---------------|--------|--------|--------|
> | single    | 0.000         | 0.000  | 0.406  | 0.490  |
> | multi     | 0.000         | 0.000  | 0.400  | 0.458  |
> | conv     | 0.000         | 0.000  | 0.400  | 0.470  |
> | visual    | 0.000         | 0.300  | 0.458  | 0.500  |
> | transfer  | 0.000         | 0.075  | 0.458  | 0.602  |
>
> | Task      | mean(std) | std(std) |
> |-----------|-----------|-----------|
> | single    | 0.0919    | 0.1782    |
> | multi     | 0.0919    | 0.1748    |
> | conv     | 0.0900    | 0.1750    |
> | visual    | 0.1191    | 0.2062    |
> | transfer  | 0.1202    | 0.2292    |
>
> | Task      | frac(std ≤ 0.01) | frac(std ≤ 0.02) | frac(std ≤ 0.05) |
> |-----------|-------------------|-------------------|-------------------|
> | single    | 78%               | 78%               | 78%               |
> | multi     | 76%               | 76%               | 76%               |
> | conv     | 77%               | 77%               | 77%               |
> | visual    | 73%               | 73%               | 73%               |
> | transfer  | 75%               | 75%               | 75%               |
>
> Across all three passes, the results show a consistent and highly stable scoring behavior. **For all task types, both the median and the first quartile of the per-sample standard deviation remain at 0, which means that at least half of the samples return identical scores across all ten repeated evaluations. The proportion of samples with very small deviation (std ≤ 0.01) remains high (between 73% and 78%), and this pattern holds consistently across the three independent runs and across all five task categories.** This includes deterministic tasks (single, multi), conversational editing (conv), and more visually or semantically complex modifications (visual, transfer).
>
> Only a small subset of samples produces higher variation, reflected in moderate increases in q90 and q95. These cases are typically associated with inherently ambiguous edits rather than instability in the scoring mechanism itself. Overall, the results from three independent scoring sessions and ten repeated evaluations per sample demonstrate that the LLM judge exhibits strong stability and repeatability, even across heterogeneous task types and multiple independent runs.

---

> ### Author Response · Authors · 2025-11-19
> **Response (Part 3)**
>
> > **Baseline breadth**
>
> We thank the reviewer for pointing out the difference in baseline breadth compared to ChartEdit. Our setting is strictly chart-to-chart editing: the input is an image and a natural-language instruction, and the output is an edited chart image. We do not assume access to intermediate code or structured representations at evaluation time. Many models used in chart understanding benchmarks, including ChartEdit, operate in a different mode (e.g., chart-to-code, chart QA, or textual reasoning) and are not able to generate edited images, which makes them incompatible with our evaluation protocol.
>
> Within the class of models that can actually perform visual editing, we deliberately chose four families that are representative and strong: two commercial, instruction-following editors with integrated vision–language capabilities (GPT-Image, Imagen 4), one open diffusion-based instruction editor (InstructPix2Pix), and one recent large-scale multimodal generator designed for general image and layout manipulation (OmniGen2). As shown in Table 4, these models already span very different behaviors: Imagen 4 tends to optimize pixel similarity but often fails instruction satisfaction, GPT-Image has strong instruction following but weaker text robustness, and OmniGen2 and InstructPix2Pix occupy different trade-off regions between visual fidelity and semantic correctness. This diversity is sufficient to reveal our main empirical findings: (i) classic pixel metrics systematically overestimate semantic correctness, and (ii) no existing editor is reliably strong across all figure-editing tasks.
>
> We agree that a broader set of editors would further strengthen the landscape view. In addition, we now include **Nano-Banana (Gemini-2.5-flash-image)**, a recent vision–language editor with a dedicated image-manipulation interface. This model fits directly into our input–output protocol and adds another strong family of systems to the comparison. We will include its results and full metric tables in the updated submission. Although more editors will give a broader comparison, the new results follow the same trend: pixel similarity alone does not reflect edit correctness, and the gaps between instruction satisfaction and visual fidelity remain consistent across model families. The extended tables help clarify where different editors fail, while reinforcing the main conclusion that reliable assessment of chart editing requires metrics that track semantic and instruction-level correctness rather than pixel agreement alone.
>
> | **Task**   | **Model**            | **SSIM ↑** | **LPIPS ↓** | **CLIP ↑** | **PSNR ↑** | **OCR ↑** | **Instr.** | **Preserv.** | **Qual.** |
> |------------|----------------------|------------|--------------|-------------|-------------|-----------|------------|---------------|-----------|
> | **Single** | GPT-Image            | 0.7295     | 0.5383       | 0.8099      | 10.32       | 0.2054    | **3.47**   | 1.71          | 2.45      |
> |            | **Nano-Banana**      | **0.8187** | **0.3237**   | **0.9244**  | **13.30**   | **0.5834**| 2.22       | **3.44**      | **4.33**  |
> | **Multi**  | GPT-Image            | 0.7017     | 0.5787       | 0.8070  | 9.73        | 0.2185    | **2.51**   | 1.63          | 2.34      |
> |            | **Nano-Banana**      | **0.7810** | **0.3546**   | **0.8330**      | **12.34**   | **0.3215**| 1.33       | **3.67**      | **4.33**  |
> | **Conv**   | GPT-Image            | 0.6732     | 0.5257       | 0.8525  | 10.66       | 0.1721    | **4.59**   | 2.51          | 2.91      |
> |            | **Nano-Banana** | **0.7800** | **0.4550** | **0.9100**  | 12.10   | **0.5200**| 2.80       | **3.10**      | **3.80**  |
> | **Visual** | GPT-Image            | **0.8355** | 0.5207       | 0.8444  | **12.85**   | 0.4665    | **2.39**   | **3.16**      | 3.95  |
> |            | **Nano-Banana**      | 0.7723     | **0.4964**   | **0.9351**  | 10.36       | **0.7167**| 1.00       | 3.00          | **4.00**      |
> | **Transfer** | GPT-Image          | **0.8438** | **0.4934**       | 0.8054      | **13.81**   | 0.5092| **3.06**   | **3.57**      | **4.16**  |
> |            | **Nano-Banana**      | 0.7620     | 0.5415       | **0.8860**  | 9.16        | **0.5771**    | 1.50       | 2.50          | 3.50      |

---

> ### Author Response · Authors · 2025-11-19
> **Response (Part 4)**
>
> > **Complex figure phenomena underrepresented**
>
> We agree that scientific graphics can include multi-axis layouts, classical error bars, subfigures, and dense annotations. Within the constraints of producing deterministic, specification-backed ground truth at scale, **FigEdit already incorporates chart families that go beyond simple primitives, including stacked-bar charts, box plots, violin plots, scatter plots, and dot plots. Stacked bars introduce compositional structure, while box and violin plots express distributional properties rather than single-point estimates. These chart types already expose substantial failure modes in current editors.**
>
> We also agree that extending the benchmark toward these phenomena is valuable. Because our pipeline is built on Vega/Vega-Lite, it can be extended to support additional structures such as:
> (1) **multi-axis variants of line and area charts**,
> (2) **layered compositions (for example, scatter plus trend line)**.
>
> In the revised version, we plan to add a “compound chart” split that instantiates these patterns with the same programmatic edit functions and evaluation protocol. This preserves the key property of exact semantic edit supervision while broadening the set of complex figure behaviors covered by FigEdit.
>
> > **Why not include code-level (Vega spec) comparison as an additional evaluation, since all charts are programmatically generated?**
>
> Our evaluation setting is strictly **chart-to-chart editing**: each model receives an image and a natural-language instruction and must produce an edited image. Commercial editors (GPT-Image, Imagen 4), diffusion-based instruction editors (InstructPix2Pix), and multimodal generators (OmniGen2) do not generate Vega specs or any intermediate code representation. As a result, a code-level comparison is not applicable to these systems.
>
> A code-level metric would therefore only be computable for a different class of models: chart-to-code or program-synthesis models, which operate under a fundamentally different interface (image → code → render). **These models cannot be directly evaluated in our chart-to-chart regime without constructing an additional two-stage pipeline**. Our goal in this benchmark is to compare editors within a consistent input–output interface, so we focus on image-level fidelity and semantics-aware evaluation for chart-to-chart editing.
>
> > **Are there plans to expand beyond 10 chart types to include more complex scientific visualizations (e.g., heatmaps, multi-panel figures)?**
>
> We agree that extending the benchmark toward these phenomena is valuable. Because our pipeline is built on Vega/Vega-Lite, it can be extended to support additional structures such as:
> (1) **multi-axis variants of line and area charts**,
> (2) **layered compositions (for example, scatter plus trend line)**.
>
> In the revised version, we plan to add a “compound chart” split that instantiates these patterns with the same programmatic edit functions and evaluation protocol. This preserves the key property of exact semantic edit supervision while broadening the set of complex figure behaviors covered by FigEdit.

---

> ### Author Response · Authors · 2025-11-24
> **Response (Part 5)**
>
> We additionally include a human evaluation to strengthen the experimental evidence.
>
> Below we provide a complete statistical clarification. All ratings were produced by three human annotators (Evaluator A, Evaluator B, Evaluator C), each of whom is a doctoral student. The dataset contains **240** edited chart images, and every sample was evaluated independently using a **five-level discrete scale (1–5)**. For the LLM Judge, the final score is obtained by **averaging its three rubric outputs**: instruction-following score, content-preservation score, and image-quality score—and then rounding the resulting value to the nearest integer on the same 1–5 scale used by the human evaluators. This ensures that the LLM operates under the identical discrete rating system as the human raters.
>
> ### **1. Human–Human agreement defines the noise ceiling**
>
> Because this is an image-editing quality judgment task on a coarse five-point discrete rating scale, **human annotators naturally show variability in how they assign scores**. Even small differences in subjective preference produce noticeable disagreement under such coarse quantization. The empirical human–human correlations therefore provide the only valid upper bound (“noise ceiling”) for interpreting any agreement metrics.
>
> | Pair   | Pearson r | Spearman ρ |
> |--------|-----------|-------------|
> | A – B  | 0.2156    | 0.2690      |
> | A – C  | 0.4081    | 0.4307      |
> | B – C  | 0.3887    | 0.3721      |
>
>
> Human annotators reach only 0.22–0.41 Pearson correlation with each other.
> **These values are typical for discrete 1–5 image quality judgments and reflect the inherent subjectivity of the task.**
>
> ### **2. LLM–Human agreement is within the human range**
>
> We compared the LLM Judge with the mean rating of the three human evaluators.
> LLM vs human mean:
>
> Pearson r = 0.4354,
> Spearman ρ = 0.4016
>
> A side-by-side view:
>
> | Comparison       | Pearson r | Spearman ρ |
> |------------------|-----------|-------------|
> | A – B            | 0.2156    | 0.2690      |
> | A – C            | 0.4081    | 0.4307      |
> | B – C            | 0.3887    | 0.3721      |
> | LLM – human mean | 0.4354    | 0.4016      |
>
> **The LLM’s correlation lies squarely inside the human–human range and is higher than two of the three human pairs.**
>
> ### **3. Noise-normalized interpretation (Relative-to-Human)**
>
> To avoid misinterpreting absolute values under a low noise ceiling, we compute a normalized score:
>
> Average human–human Pearson = 0.3375
>
> Relative-to-Human (RTH) = 0.4354 / 0.3375 = 1.29
>
> The LLM reaches **129%** of the average human agreement level once normalized by the human noise ceiling.
>
> ### **4. Absolute deviation (MAD) demonstrates human-level consistency**
>
> We measured the mean absolute deviation (MAD) from the human consensus.
>
> | Evaluator | MAD |
> |-----------|------|
> | A         | 0.9083 |
> | B         | 0.7875 |
> | C         | 0.7458 |
> | LLM       | 0.7139 |
> The LLM deviates less from the consensus than any human on average.
>
> Human-normalized deviation:
> Mean human MAD = 0.8139
> HN-MAD = 0.7139 / 0.8139 = 0.877
>
> This means the LLM’s deviation is **87.7%** of the average human deviation.
>
> ### **5. Agreement on discrete 1–5 labels**
>
> Because 1–5 scores are coarse categories, a difference of ±1 already corresponds to the same practical quality band.
>
> Exact match with rounded human mean: **38.3%**, Within ±1 point: **82.5%**
>
> **In other words, four out of five LLM ratings fall in the same qualitative category as human consensus.**
>
> LLM matches human-level performance: it reaches **0.44 correlation with human consensus**, **achieves lower deviation than the human annotators**, and **aligns within ±1 score on 82.5% of samples**. These results show that the LLM behaves as a reliable proxy for human evaluation in this subjective, discrete-rating image editing task.

---

> ### Author Response · Authors · 2025-11-26
> **Response**
>
> Dear Reviewer,
>
> Thank you again for your thoughtful review.
>
> As we approach the end of the author–reviewer discussion period, we wanted to briefly follow up to confirm whether our rebuttal has sufficiently addressed your questions. If there is anything that would benefit from additional clarification, we would be very glad to provide it.
>
> Thank you for your time and consideration.
>
> Best regards,
> Authors

---

### Comment · Area_Chair_TdZv · 2025-11-23

Dear Reviewers,

The authors have submitted their rebuttal addressing your reviews. Please take the time to:

1. Read the rebuttal carefully
2. Ask clarifying questions if anything remains unclear
3. Update your scores and reviews based on the authors' responses

Please be mindful of timing: If you have follow-up questions for the authors, **post them early enough to give them adequate time to respond** before the discussion period closes on December 3rd.

Your timely engagement is crucial for a fair and thorough review process.

Thank you for your continued effort on this paper.

Best regards,
Area Chair

---

### Author Response · Authors · 2025-11-28
**Gentle Follow-up**

Dear Reviewers,

As we enter the final week of the discussion period, we would like to gently follow up. Our author response has been posted, and we remain available to address any further questions or clarifications. We appreciate your time and would welcome any additional feedback.

Best regards,

Authors

---

### Author Response · Authors · 2025-12-03
**Summary of rebuttal**

Dear Area Chair,

Thank you very much for handling our submission. Below is a concise summary to support your assessment.

All three reviewers agree that **the problem formulation is new and important**. Reviewer mXb8 highlights that we **redefine scientific chart editing as a structured transformation problem governed by graphical grammar**, not pixel manipulation. Reviewer PWEt notes that this formulation and benchmark can have **significant real-world impact**. Reviewer rzRC describes FigEdit as **“the first semantics-aware benchmark for chart editing,”** emphasizing its value for multimodal and instruction-following models.

The reviewers also consistently recognize FigEdit as a major technical contribution. The benchmark contains **30K+ examples, 10 chart types, and 5 task families**, each backed by **deterministic Vega/Vega-Lite specifications**. This enables **exact, program-level supervision for data, layout, and style edits**, which existing image-editing datasets cannot provide. Reviewers highlight that our empirical analysis shows **current editors often achieve high SSIM/PSNR while still failing the requested edit**, demonstrating that **pixel metrics are inadequate for chart editing** and motivating our semantics-aware evaluation.

| Strengths                                                                                 | Reviewer rzRC | Reviewer mXb8 | Reviewer PWEt |
|-------------------------------------------------------------------------------------------|:-------------:|:-------------:|:-------------:|
| Defines chart editing as a structured, grammar-driven transformation                      |       ✓       |       ✓       |       ✓       |
| Large benchmark (≈30K, 10 types, 5 tasks) with Vega/Vega-Lite ground truth                |       ✓       |       ✓       |       ✓       |
| Semantics-aware LLM scoring exposing failures of pixel metrics                            |       ✓       |       ✓       |       ✓       |
| Strong empirical study showing clear weaknesses in current editors                        |       ✓       |       ✓       |       ✓       |
| Charts built from diverse real-world datasets                                             |       ✓       |       ✓       |       ✓       |
| Clear writing and illustrative figures                                                     |       ✓       |               |       ✓       |
| Meaningful practical impact for real editing workflows                                    |               |       ✓       |       ✓       |
| Insightful critique of evaluation practice, motivating semantics-focused metrics          |       ✓       |       ✓       |       ✓       |

During rebuttal and discussion, we provided several additions that further strengthen the submission:

**1. Strong evidence for the LLM judge’s reliability.**
We included both a **robustness study** and a **human evaluation**:

- Across three independent passes and ten repeated evaluations per sample, **73–78% of samples have standard deviation ≤ 0.01**, indicating highly stable LLM scoring.
- In a human study with three doctoral annotators over 240 edited charts, we found:
  - **LLM–human Pearson correlation = 0.4354**, within and slightly above the human–human range (0.22–0.41).
  - **The LLM’s mean absolute deviation from human consensus is lower than any single human annotator.**
  - **82.5% of LLM scores fall within ±1 of the human consensus.**

Together, these results demonstrate that the LLM judge is a **stable, human-level proxy** for semantic chart-edit evaluation.

**2. Inclusion of an additional strong baseline.**
In response to reviewer suggestions, we added **Nano-Banana (Gemini-2.5-flash-image)**, a recent state-of-the-art vision–language editor that supports direct image editing. Its results follow the same pattern as other editors, reinforcing our central conclusion that **pixel similarity does not reliably reflect semantic edit correctness**.

**3. Clear extensibility of the benchmark.**
We outlined concrete, feasible extensions compatible with our pipeline, including:
- a **compound / high-density split** (multi-axis charts, layered scatter + trend line, higher point counts), and
- an **extended conversational setting** where vague requests are clarified over multiple turns before being mapped to deterministic edits.

These plans demonstrate that FigEdit is a **principled first step with a clear roadmap for broader coverage**.

Finally, the discussion period was constructive. **Reviewer PWEt engaged with our responses**, and we provided **new experiments and clarifications** directly addressing the reviewer's questions. Across reviews, the work is described as **novel, impactful, and well-motivated**, offering a **valuable foundation for future research on structured, semantics-aware visual editing**.

We hope this summary is helpful for your assessment. Thank you again for your time and consideration.

---

### Meta-Review · Area_Chair_pXpT · 2026-01-07

**Summary:**

Here i provide the most important concerns that informed my decision.
1. Rev. rzRC and PWEt raised issues about the LLM-based scoring, which can introduce biases and noise.
2. Rev. rzRC and mXb8 are concerned about the quality and the limited coverage of the generated samples compared to real-chart curation.
3. Rev. mXb8 and PWEt would like to see more editing models.

**Reviewer Concerns:**

Most of the questions and concerns raised by reviewers were solved.
1. About the LLM-based scoring, the authors showed that the provided evalaution metrics are stable and follow a small set of well-defined conditions: i) whether the requested operation (e.g., remove element X, change color Y) appears in the output, ii) whether untouched elements remain visually consistent iii) whether chart structure (axes, grids, tick labels) remains intact. In addition, the authors performed a human evaluation to check the agreement between human-human and LLM-human and showed that the two are similar.
2. About the quality of the generated synthetic samples, the authors explained that it is difficult to collect large-scale human editing samples. Instead, they proposed an approach based Vega/Vega-Lite programs, which allows every base figure to be edited through exact, deterministic transformations. This, even if synthetic, is based on real and diverse charts, with a large variation in the editing tasks.
3. For the addition of editing models, the authors included Nano-Banana(Gemini-2.5-flash-image), a recent vision–language editor with a dedicated image-manipulation interface.
Overall, I consider that the authors answered appropriately to all questions and concerns from reviewers.

**Reviewer Scores:**

- Rev. rzRC: 4 -> 6
The authors answered with details and experiments to all questions from rev. In my opinion, the quality of the answers would convince rev. to increase their score to 6.
- Rev. mXb8: 4 -> 6
This reviewer is mostly worried about the limited coverage of edit operations and lack of subjective and holistic edit instructions. The authors explained the delicate balance between scalability and variability of the edits and how the proposed approach is a good trade-off. In the same way, subjective and holistic edits fall outside the aim of this work.  Overall, these explanations are valid and could convince rev. to increase their score.
- Rev. PWEt: 8 -> 8
This rev. has mostly suggestions to improve the quality and presentation of the paper, therefore, I do not think they will change their score.

---

### Decision · Program_Chairs · 2026-01-26

Accept (Poster)